# DBRNet: Advancing Individual-Level Continuous Treatment Estimation through Disentangled and Balanced Representation

## Abstract

Estimating the individual-level continuous treatment effect holds significant practical importance in various decision-making domains, such as personalized healthcare and customized marketing. However, current methods for individual treatment effect estimation are limited to discrete treatments or rely on a simplistic approach of balancing the entire representation, which may lead to inaccurate estimation. To the best of our knowledge, no existing efforts is capable of precisely adjusting for selection bias in continuous settings. Hence, in this paper, we propose a novel Disentangled and Balanced Representation Network (DBRNet) for estimating the individualized dose-response function (IDRF), which learns disentangled representations and precisely adjusts for selection bias. Extensive results on synthetic and semi-synthetic datasets demonstrate that our DBRNet outperforms most state-of-the-art methods. Our code is avaiable at https://anonymous.4open.science/r/DBRNet_final_2-2B76.

## 1 Introduction

In various fields, from medicine to marketing, estimating the causal effects of continuous treatments at individual level is not just an academic exercise — it's crucial for decision-making. Take precision medicine as an example: the central question often focuses on determining the *"optimal dosage of medicine to achieve the optimal outcome for a given patient"*. Therefore, understanding the causal relationship between continuous treatments and the outcome can assist us in developing customized medication regimens tailored to specific patients.

In estimating the individual treatment effect (ITE), two predominant challenges arise: the inability to observe *counterfactual outcomes* and the presence of *selection bias*. For instance, when assigning a specific dosage of treatment to a patient, only the factual outcome corresponding to that dosage is observed, leaving other potential counterfactual outcomes for alternate dosages unobserved. Moreover, unlike in Randomized Controlled Trials where treatments are assigned at random, the dosage a patient receives in practice may depend on certain patient-specific features (e.g., older individuals may more frequently receive higher dosages). This dependency can introduce selection bias, thereby compromising the accuracy of counterfactual outcome estimations. For example, it becomes challenging to accurately estimate the treatment effect of higher dosages in younger populations. Beyond conventional causal inference techniques such as stratification methods (Pearl, 2009) and matching methods (Abadie et al., 2004), recent research has harnessed representation learning for counterfactual prediction and selection bias mitigation (Johansson et al., 2016; Shalit et al., 2017; Schwab et al., 2020; Bellot et al., 2022; Curth & van der Schaar, 2021; Wang et al., 2022). This approach involves deriving latent representations from covariates, balancing these representations across treatment groups to eliminate selection bias, and subsequently estimating counterfactuals on these balanced representations (Shalit et al., 2017).

Despite prior endeavors, several crucial challenges still remain unresolved. ❶ Most existing studies are limited to discrete treatment settings. These methods cannot be easily extended to continuous treatment settings due to the infinite number of unobserved counterfactual outcomes and the difficulty of adjusting for selection bias in an infinite set of treatments. ❷ To adjust for selection bias, existing methods resort to a simple and brutal approach of balancing the entire representation. However, we

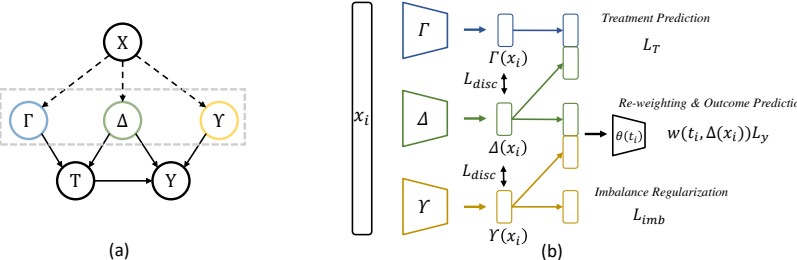

Figure 1: Causal graph and framework of our DBRNet. Figure (a) shows the causal graph involving covariates ($X$), treatment ($T$), outcome ($Y$), instrumental factors ($\Gamma$), confounder factors ($\Delta$), adjustment factors ($\Upsilon$). The solid line represents causal relations, and dot lines denote affiliations. Figure (b) shows the framework of DBRNet. Three contracted neural networks are utilized to obtain the deep representations of the three factors. Then $\Gamma(x_i)$ and $\Delta(x_i)$ are concatenated to predict the distribution of $t_i$. $\Delta(x_i)$, and $\Upsilon(x_i)$ are used to predict outcomes through a varying coefficient network $g_{\theta(t_i)}$, while $\Upsilon(x_i)$ attempts to encode little information about treatment.

argue that not all information in the latent representation should be balanced to adjust for selection bias. For example, although confounder factors in the representation bring selection bias, they also contribute to outcome predictions. Balancing instrumental factors in the representation is also theoretically implausible since they are related to the treatment assignment and should not be identical across treatments. Therefore, the entire representation of input covariates should not be indiscriminately balanced.

Although some follow-up methods have attempted to address the first challenge by dividing continuous treatments into bins and assigning one network head to each bin (Schwab et al., 2020), using a varying coefficient structure (Nie et al., 2021), or designing a specific loss for continuous treatments (Bellot et al., 2022), these methods still cannot ensure precise adjustment for selection bias (i.e., the second challenge). Hassanpour & Greiner (2019b) leveraged disentangled representations to distinctly embed instrumental, confounder, and adjustment factors, thus eliminating selection bias and estimating ITE based on their proposed causal graph. Although the model can precisely correct for selection bias, it is based on the binary treatment assumption and cannot be readily extended to continuous settings (i.e., the first challenge). To the best of our knowledge, there is no research that simultaneously solves these two problems, namely *generating appropriate disentangled representations that precisely adjust for selection bias to estimate the continuous treatment effect at the individual level, which is defined as the Individualized Dose-Response Function (IDRF).*

To address these challenges, we propose a novel method named Disentangled and Balanced Representation Network (DBRNet). Specifically, we assume that covariates are determined by three latent factors: (1) Instrumental factors (2) Confounder factors (3) Adjustment factors (as shown in Fig. 1 A). DBRNet first learns disentangled representations for each factor, providing the opportunity to precisely adjust for bias by using only the relevant representations instead of the entire representation. Then, we precisely adjust for selection bias by adopting a re-weighting function and predict outcomes based on the representations of confounder and adjustment factors through a varying coefficient network, which enables continuous treatment effect estimation. A rigorous theoretical proof supporting the debiasing of our re-weighting function is also furnished. Our contributions are as follows.

- We propose a new method, DBRNet, which learns disentangled and balanced representations for continuous treatment effect estimation at the individual level.

- To the best of our knowledge, DBRNet is the first model to precisely adjust for selection bias in continuous treatment settings substantiated by theoretical proofs.

- We have conducted extensive experiments to validate the effectiveness and the disentangling ability of our model. The results show that our method achieves promising results on both synthetic and semi-synthetic datasets.

## 2 RELATED WORK

As mentioned above, selection bias and counterfactual prediction are two major challenges for ITE. To address these issues, various methods have been proposed (Stuart, 2010; Yao et al., 2021; Hansen,

2008; Chipman et al., 2010; Hansen, 2008; Wager & Athey, 2018). Moreover, several state-of-the-art methods use deep representation learning models to estimate ITE based on treatment-invariant representations (Johansson et al., 2016; Shalit et al., 2017; Chu et al., 2020; Yao et al., 2018; Bellot et al., 2022; Wang et al., 2022). Specifically, the discrepancy loss between the deep representations of the treatment group and control group is used to balance the distribution of the two groups to adjust for selection bias. Subsequently, one network head for each treatment is built on the deep balanced representations to estimate ITE. However, due to the commonly used two-head design in existing work, these models cannot be easily generalized to continuous treatment settings.

To address this limitation and estimate the average dose-response function (ADRF) under a continuous setting, several methods have been proposed (Schwab et al., 2020; Nie et al., 2021; Shi et al., 2019; Bellot et al., 2022). Schwab et al. (Schwab et al., 2020) introduced a modification of TARNET (Shalit et al., 2017), called Dose Response Networks (DRNet), which divided continuous dosage into equally sized strata and assigned a head to each of them. To further achieve continuity of the ADRF, Nie et al. (Nie et al., 2021) proposed a varying coefficient neural network (VCNet). Instead of using a multihead model, they used a varying coefficient prediction head whose weight depends on the treatment $t$, which improves the expressiveness of the treatment effect. In addition, generative adversarial net (GAN)-based models (Bica et al., 2020) and transformer-based models (Zhang et al., 2022) have also been proposed to estimate the ADRF. However, the methods fail to accurately adjust for selection bias, as they do not make any adjustments to eliminate the bias, or resort to a simple and unsophisticated approach to balance the entire representation.

Hassanpour & Greiner (2019a) proposed that disentangled representations of covariates enhance the capture of their underlying factors, thereby improving ITE estimation performance. Instead of balancing the entire deep representation, they only applied discrepancy loss on the representation of adjustment factors and used re-weighting methods to adjust for selection bias brought by confounders. As a result, no confounding variables are discarded. However, the method can only work with binary treatments. While subsequent research efforts (Curth & van der Schaar, 2021; Chauhan et al., 2023) have introduced more refined disentanglement representations, it is worth noting that these advancements are predominantly tailored to binary settings.

Unlike existing work, our DBRNet attempts to address the two limitations mentioned above by generating appropriate disentangled representations for three underlying factors and precisely adjusting for selection bias to estimate IDRF.

## 3 METHODOLOGY

In this section, we begin by defining the problem and presenting an overview of the structure of DBRNet. Next, we delve into the functions and explanations of each component of our DBRNet. Following this, we provide theoretical substantiation that our devised re-weighting function is able to precisely adjust for selection bias. Lastly, we discuss the advantages of our approach in comparison to other state-of-the-art models.

### 3.1 PROBLEM SETTING

Let $\mathcal{D} = \{x_i, y_i, t_i\}_{i=1}^N$ denote a dataset of size $N$, where $x_i, y_i, t_i$ are independent realisations of random variables $X, Y, T$ with support $(\mathcal{X}, \mathcal{Y}, \mathcal{T} = [0, 1])$, respectively. We refer to $X \in \mathbb{R}^m$ as covariates, which contain information about features of an unit. $Y$ represents the outcome, and $T$ represents a continuous treatment in the range from 0 to 1. Our goal is to estimate the Individualised Dose-Response Function (IDRF) with continuous treatments by eliminating selection bias.

**Definition 1** *(Individualised Dose-Response Function (IDRF)). IDRF measures the treatment effect for an individual under continuous treatments, which can be defined as:*

$$\mu(t, x) = \mathbb{E}[Y(T = t)|X = x, T = t] \tag{1}$$

We assume that covariates are generated from three underlying factors: (1) *instrumental factors* ($\Gamma(x)$) that are associated with the treatment but not with the outcome except through the treatment, (2) *confounder factors* ($\Delta(x)$) that are associated with both the treatment and outcome, and (3) *adjustment factors* ($\Upsilon(x)$) that are predictive of the outcome but not associated with the treatment.

Therefore, the treatment assignment is affected by instrumental factors and confounder factors, while the outcome is affected by confounder factors and adjustment factors. This assumption is a commonly used approach in previous research to disentangle the covariates for precise information extraction (Yao et al., 2021; Wu et al., 2020), an example can be found in Appendix H. The underlying relationship can be illustrated using a causal graph shown in Fig. 1(a). It is important to note that IDRF identification is made under the following assumptions.

**Assumption 1 (Stable Unit Treatment Value Assumption (SUTVA))** *There are no interactions between units, and there is only one version of each treatment, that is, different levels or doses of one treatment are treated as different treatments.*

**Assumption 2 (Ignorability)** *The potential outcome $Y(T = t)$ is independent of the treatment assignment given all covariates. Formally, $Y(T = t) \perp\!\!\!\perp t | X$.*

**Assumption 3 (Positivity)** *Every unit should have non-zero probabilities to be assigned in each treatment group. Formally, $\mathbb{P}(T = t | X = x) \neq 0, \forall t \in \mathcal{T}, \forall x \in X$.*

**Assumption 4 (Generation of Covariates)** *Given a set of covariates, denoted as $X$, we assume $X$ follows the joint distribution of instrumental factors $\Gamma$, confounder factors $\Delta$, and adjustment factors $\Upsilon$. Formally, $\mathbb{P}(X) = \mathbb{P}(\Gamma, \Delta, \Upsilon)$.*

## 3.2 DBRNET MODEL

DBRNet is designed according to the causal graph in Fig. 1. It first learns disentangled representations of covariates and further corrects for selection bias using relevant representations rather than the entire ones. Three contracted feedforward neural networks are utilized to obtain disentangled representations of three factors $\{\Gamma(x_i), \Delta(x_i), \Upsilon(x_i)\}$ defined in Section 3.1. Then $\Gamma(x_i)$ and $\Delta(x_i)$ are concatenated to predict the distribution of $t_i$ using a conditional density estimator $\mathbb{P}(t_i | \Gamma(x_i), \Delta(x_i))$. $\Delta(x_i)$ and $\Upsilon(x_i)$ are used to predict the final outcome through a varying coefficient network $g_{\theta(t_i)}(\Delta(x_i), \Upsilon(x_i))$, while $\Upsilon(x_i)$ attempts to encode little information about treatment. Typically, a re-weighting function is responsible for precisely adjusting for selection bias. The framework of DBRNet is shown in Fig. 1(b).

The objective function of DBRNet is as follows:

$$J(X, T, Y) = \frac{1}{N} \sum_{i=1}^{N} (w(t_i, \Gamma(x_i), \Delta(x_i)) L_y + \alpha L_T + \beta L_{disc} + \gamma L_{ind} + \lambda L_{reg}), \quad (2)$$

where $w(t_i, \Gamma(x_i), \Delta(x_i))$ denotes the re-weighting function to mitigate selection bias; $L_y$ and $L_T$ are the prediction losses for outcome $Y$ and treatment $T$, respectively. $L_{disc}$ quantifies discrepancies between latent representations $(\Gamma(x_i), \Delta(x_i), \Upsilon(x_i))$; $L_{ind}$ promotes the independence between $\Upsilon(x_i)$ and treatment $t_i$; while $L_{reg}$ serves as a regularization term against overfitting. $\alpha$, $\beta$, $\gamma$, and $\lambda$ are hyperparameters balancing the different terms in the objective function. In the following, we present the details of each term.

**Factual Loss.** Factual loss is used to force $\Delta(x_i)$ and $\Upsilon(x_i)$ to extract more predictive information from the covariates. As in Fig. 1(a), we aim to estimate the factual outcome $y_i$ from $\Delta(x_i), \Upsilon(x_i)$, and assigned treatment $t_i$ for unit $i$. To simultaneously preserve the influence of treatment and maintain the continuity of the dose-response curve, we adopt a varying coefficient neural network $g_{\theta(t_i)}$ to estimate outcomes (Nie et al., 2021). Factual loss is computed by comparing the ground truth $y_i$ with our estimated value:

$$L_y = L\left(y_i, g_{\theta(t_i)}(\Delta(x_i), \Upsilon(x_i))\right). \quad (3)$$

The varying coefficient structure utilizes a function $g_{\theta(t)}$ with varying parameters $\theta(t)$ instead of fixed parameters to predict outcomes. By leveraging this structure, continuous treatment effect can be estimated by incorporating continuous $t$ in outcome estimation. Especially, a B-spline of degree $p$ with $q$ knots, resulting in $k = p + q + 1$ basis, is used to model $\theta(t)$. Let $\mathbf{B} = [b_1, b_2, ..., b_k] \in \mathbb{R}^{n \times k}$ denote the spline basis for the treatment $T \in \mathbb{R}^{n \times 1}$. For a single-layer feedforward network with $p$ inputs and $q$ outputs, the function is given by $f_{\theta(t)} = \sum_{i=1}^{k} (b_i \cdot (\mathbf{XW}))$, where $\mathbf{W} \in \mathbb{R}^{p \times q \times k}$ is the optimizable weight.

**Treatment Loss.** We incorporate a treatment loss in our model to enhance the encoding of $\Gamma(x_i)$ and $\Delta(x_i)$ with respect to $t_i$. Prior studies have often predicted the probability of treatment using the full covariate representation (Shi et al., 2019; Nie et al., 2021). This approach can inadvertently leverage irrelevant information, such as adjustment factors, from the representation. To mitigate this, we estimate the probability of $t_i$ from the concatenation of $\Gamma(x_i)$ and $\Delta(x_i)$ using a conditional density estimator $\pi(t|\Gamma(x_i), \Delta(x_i))$, yielding more accurate treatment predictions. In this paper we adopt a naive density estimator (Nie et al., 2021), which approximates the conditional density by dividing $t \in [0, 1]$ equally into $B$ grids, and estimating the conditional density $\pi(t|\Gamma(x_i), \Delta(x_i))$ on the $B+1$ grid points using a simple neural network $\pi^{NN}(\Gamma(x_i), \Delta(x_i)) = softmax(w, \Gamma(x_i), \Delta(x_i)) \in \mathbb{R}^{B+1}$, and the densities for other values of $t$ are derived via linear interpolation. Performance is measured using a negative logarithmic likelihood loss.

$$L_T = -\log[\mathbb{P}(t_i|\Gamma(x_i), \Delta(x_i))]. \tag{4}$$

**Discrepancy Loss.** Discrepancy loss is applied to ensure the independence of the three representations, namely $\Gamma(x_i)$, $\Delta(x_i)$, and $\Upsilon(x_i)$, and thus strengthen disentanglement in the latent space. The formula for this loss is presented as follows:

$$L_{disc} = \frac{1}{L_D(\Gamma(x_i); \Delta(x_i)) + L_D(\Delta(x_i); \Upsilon(x_i))}, \tag{5}$$

, where $L_D(\Gamma(x_i); \Delta(x_i))$ denotes the divergence loss, inspired by the KL divergence, between $\Gamma(x_i)$ and $\Delta(x_i)$. If $\Gamma(x_i)$ and $\Delta(x_i)$ are identical, the result will be 0. Conversely, if $\Gamma(x_i)$ and $\Delta(x_i)$ are distinctly different, the value will be larger. Therefore, through the collaboration of other modules, $L_{disc}$ encourages all disentangled representations to encode only the relevant information. The definition of $L_D$ and additional details can be found in Appendix D

**Independent Loss.** Independent loss is applied to ensure that the learned factors $\Upsilon(x_i)$ do not embed any information about $t_i$ and that all information related to $t_i$ is encoded in $\Gamma(x_i)$ and $\Delta(x_i)$. Previous studies (Shalit et al., 2017; Johansson et al., 2016; Hassanpour & Greiner, 2019a;b) have emphasized the need to balance adjustment representations for treatment group and control group (binary treatment setting), which aims to induce the desired independence using a discrepancy loss. However, within the framework of continuous treatments, the endeavor to achieve balance adjustment representations for every value $t_i$ becomes infeasible. Hence, we intend to push $\Upsilon(x_i)$ to embed little information about the treatment by forcing the performance of the treatment probability estimation from adjustment representation to be poor, which motivates us to minimize the following "positive" log-likelihood loss:

$$L_{ind} = \log(\mathbb{P}(t_i|\Upsilon(x_i)). \tag{6}$$

In particular, this independent loss allows us to encode all information of $t_i$ in $\Gamma(x_i)$ and $\Delta(x_i)$ instead of $\Upsilon(x_i)$, which facilitates the precisely adjustment for selection bias through the re-weighting function (discussed in next paragraph). This is one of the key contributions of our paper. Furthermore, compared to previous studies that balance the entire representation of covariates, our approach does not discard confounder factors since they contain valuable information about outcome prediction (Hassanpour & Greiner, 2019b). Moreover, we also exclude instrumental factors since they should not be balanced according to the causal theory.

**Re-weighting Function.** Recall one of our objectives is to precisely eliminate selection bias. Inspired by (Imbens, 2000; Kloek & Van Dijk, 1978), we derive "propensity score" $\mathbb{P}(t_i|\Gamma(x_i), \Delta(x_i))$ from instrumental and confounder factors and use the inverse of it to re-weight the prediction loss of outcomes as follows:

$$w(t_i, \Gamma(x_i), \Delta(x_i)) = \frac{1}{\mathbb{P}(t_i|\Gamma(x_i), \Delta(x_i))}, \tag{7}$$

where $\mathbb{P}(t_i|\Gamma(x_i), \Delta(x_i))$ is the output of the conditional density estimator for treatment. Notably, it does not require additional computation or prior knowledge about the treatment distribution as in (Hassanpour & Greiner, 2019b). Furthermore, we can precisely remove bias attributable to the confounder and instrumental factors instead of the unrelated part ($\Upsilon(x_i)$). Detailed proof follows.

| Method | Synthetic Data | | News | | IHDP | |
|---|---|---|---|---|---|---|
| | MISE | AMSE | MISE | AMSE | MISE | AMSE |
| Dragonet | $0.1854 \pm 0.0232$ | $0.0415 \pm 0.0081$ | $1.3241 \pm 0.1617$ | $0.0535 \pm 0.0053$ | $4.7034 \pm 0.5860$ | $0.9549 \pm 0.3005$ |
| Dragonet_TR | $0.1720 \pm 0.0219$ | $0.0281 \pm 0.0095$ | $1.3147 \pm 0.1594$ | $0.0401 \pm 0.0062$ | $4.2877 \pm 0.4226$ | $0.6490 \pm 0.1660$ |
| DRNet | $0.1849 \pm 0.0232$ | $0.0409 \pm 0.0081$ | $1.3248 \pm 0.1616$ | $0.0542 \pm 0.0054$ | $4.7394 \pm 0.6036$ | $0.9581 \pm 0.3324$ |
| DRNet_TR | $0.1752 \pm 0.0334$ | $0.0315 \pm 0.0235$ | $1.3148 \pm 0.1601$ | $0.0403 \pm 0.0060$ | $4.1313 \pm 0.6320$ | $0.6140 \pm 0.1954$ |
| VCNet | $0.1545 \pm 0.0248$ | $0.0173 \pm 0.0093$ | $2.3372 \pm 0.1808$ | $0.0384 \pm 0.0367$ | $3.6651 \pm 0.6409$ | $0.6755 \pm 0.4875$ |
| VCNet_TR | $0.1418 \pm 0.0299$ | $0.0142 \pm 0.0072$ | $2.3289 \pm 0.2009$ | $0.0378 \pm 0.0401$ | $3.7935 \pm 1.3625$ | $1.2302 \pm 1.2198$ |
| TransTEE | $0.2033 \pm 0.0978$ | $0.0552 \pm 0.0884$ | $\mathbf{1.2849 \pm 0.1587}$ | $0.0153 \pm 0.0066$ | $3.3051 \pm 0.8137$ | $0.7922 \pm 0.8073$ |
| DBRNet(Ours) | $\mathbf{0.1414 \pm 0.0256}$ | $\mathbf{0.0131 \pm 0.0072}$ | $1.7846 \pm 0.2202$ | $\mathbf{0.0100 \pm 0.0075}$ | $\mathbf{3.2033 \pm 0.3550}$ | $\mathbf{0.3680 \pm 0.3300}$ |

Table 1: Performance comparison between DBRNet and baselines. Numbers reported are (MISE/AMSE± standard deviation) on Synthetic, IHDP, and News with 50 runs.

### 3.3 THEORETICAL PROOF OF BIAS ELIMINATION

In this section, we outline the derivation of our re-weighting function, inspired by importance sampling theory (Hassanpour & Greiner, 2019a; Kloek & Van Dijk, 1978). We then provide proofs for its debiasing ability.

**Definition 2** *Let $\Delta, \Upsilon : \mathcal{X} \to \mathcal{R}$ be the representation functions for confounder factors and adjustment factors respectively. Let $g_{\theta(t)} : \mathcal{R} \times \mathcal{R} \times [0,1] \to \mathcal{Y}$ be an hypothesis defined over the representation space $\mathcal{R} \times \mathcal{R}$. The expected loss for the unit and treatment pair $(x, t)$ is:*

$$l_{\Delta, \Upsilon, g_{\theta(t)}}(x, t) = \int_{\mathcal{Y}} L(Y(t), g_{\theta(t)}(\Delta(x), \Upsilon(x)))\mathbb{P}(Y(t)|x)dY(t).$$

**Definition 3** *The expected unbiased IDRF loss across all treatment $t \in T$ is:*

$$\epsilon = \mathbb{E}_x \Big[ \int_{\mathcal{T}} l_{\Delta, \Upsilon, g_{\theta(t)}}(x, t)dt \Big] = \int_{\mathcal{X}} \int_{\mathcal{T}} l_{\Delta, \Upsilon, g_{\theta(t)}}(x, t)\mathbb{P}(x)dtdx$$

**Lemma 1** *(Importance Sampling (Kloek & Van Dijk, 1978)) Let $p(x)$ be a probability density for a random variable $X$ defined on $\mathbb{R}^d$, then for any probability density $q(x) \in \mathbb{R}^d$ that satisfies $q(x) > 0$ whenever $f(x)p(x) \neq 0$, we have:*

$$\mathbb{E}_{x \sim p(x)}[f(x)] = \mathbb{E}_{x \sim q(x)}\Big[f(x)\frac{p(x)}{q(x)}\Big].$$

The Lemma suggests that importance sampling facilitates the computation of the expectation of a target function $f(x)$ under an unknown distribution $p(x)$ by weighting the function with $\frac{p(x)}{q(x)}$ under a known distribution $q(x)$.

**Theorem 1** *Let $p(x, t')$, $p(x, t)$ denote the counterfactual and factual probability density for unit $x$, respectively. Under the conditions of lemma 2, the expected loss function for counterfactual outcomes is:*

$$\epsilon_{CF} = \mathbb{E}_{x, t \sim p(x, t')}[l_{\Delta, \Upsilon, g_{\theta(t)}}(x, t)] = \mathbb{E}_{x, t \sim p(x, t)}\Big[l_{\Delta, \Upsilon, g_{\theta(t)}}(x, t)\frac{p(x, t')}{p(x, t)}\Big],$$

*where $t'$ represents all counterfactual treatments of $x$, especially $t' = \{t'|t' \in \mathcal{T}, t' \neq t\}$*

The theorem implies that the expected loss function for counterfactual outcomes can be derived by reweighting the factual loss. Hence, we can cooperate the counterfactual loss with factual loss and optimise them together through a designed weight (Hassanpour & Greiner, 2019a); $w = 1 + \frac{\mathbb{P}(x, t')}{\mathbb{P}(x, t)} = \frac{\mathbb{P}(x)}{\mathbb{P}(x, t)} = \frac{1}{\mathbb{P}(t|x)} = \frac{1}{\mathbb{P}(t|\Gamma(x), \Delta(x))}$. For further details, please refer to Appendix A.

**Theorem 2** *(Bias Removal with Weighted Loss) Let $p(x, t)$ denote the factual distribution. Under the theorem 1 and all assumptions, we have:*

$$\mathbb{E}_{x, t \sim p(x, t)}[w \cdot l_{\Delta, \Upsilon, g_{\theta(t)}}(x, t)] = \epsilon$$

This theorem states that the weighted loss is an unbiased estimation of the IDRF loss, showcasing our re-weighting function's advance in eliminating selection bias. See Appendix A for a detailed proof.

### 3.4 DISCUSSIONS

Our proposed DBRNet builds upon existing works such as (Hassanpour & Greiner, 2019b) and (Nie et al., 2021). However, our approach extends the existing works in several significant ways. (1) The disentangled method in (Hassanpour & Greiner, 2019b) can only provide a causal effect under binary treatments, while our method facilitates a mixed type of treatments, including continuous treatments. (2) We introduce a novel design for the independent loss. Instead of simply and brutally minimizing the discrepancy between the entire treatment representation and the control representation, we minimize the amount of treatment information that can be extracted from adjustment factors, which facilitates the precise adjustment for selection bias in re-weighting function. (3) We use the direct output of the conditional density estimator for re-weighting, which helps precisely eliminate bias and does not require additional computation or prior knowledge about the treatment distribution. Furthermore, in the ablation study, we demonstrate the importance of this component for model performance. (4) We provide rigorous theoretical proofs substantiating the debiasing ability of the re-weighting function. Finally, our DBRNet employs a discrepancy loss between different deep representations, which ensures that each representation only encodes the relevant information. Overall, DBRNet offers an extensive ability to generate disentangled representations on which selection bias can be precisely adjusted.

## 4 EXPERIMENTS

In this section, we present extensive experimental results to demonstrate the effectiveness of DBRNet and address the following three research questions: **Q1:** How effective is DBRNet in estimating IDRF and adjusting for selection bias compared to the state-of-the-art methods? **Q2:** What is the individual contribution of each component in our model, including the discrepancy loss, independent loss, and re-weighting function? **Q3:** Can deep representations guarantee the disentanglement of the three underlying factors?

### 4.1 DATASETS AND BASELINES

As the ground truth of treatment effects is often unknown in practice, especially for continuous treatments, existing studies mainly rely on synthetic datasets and semi-synthetic datasets that manually construct treatments and outcomes given real-world features (Curth et al., 2021; Wang et al., 2022; Bellot et al., 2022; Nie et al., 2021). We follow this convention to build one synthetic dataset and two semi-synthetic datasets: News[1] and IHDP (Hill, 2011) for evaluation. To evaluate the effectiveness of selection bias adjustment, **we intentionally design all training sets to contain selection bias, while test sets are unbiased.** Hence, if a model is trained on a biased training set and performs well on the corresponding test set, it provides evidence of the model's ability to eliminate bias. We provide detailed generating schemes following (Nie et al., 2021) for the three datasets in the Appendix F. We evaluate the performance of our proposed DBRNet against several state-of-the-art methods for IDRF estimation, including Dragonet, Dragonet_TR (Shi et al., 2019), DRNet, DRNet_TR (Schwab et al., 2020), VCNet, VCNet_TR (Nie et al., 2021), and TransEE (Zhang et al., 2022) , where TR refers to targeted regularization. We also include non-neural network based models: Causal Forest (Wager & Athey, 2018), Bart (Chipman et al., 2010) and GPS (Imbens, 2000), due to the space limitation, we show the performance in the Appendix C.

### 4.2 IMPLEMENTATION DETAILS

All implementation details and optimal hyperparameters can be found in Appendix E.

---

[1] https://archive.ics.uci.edu/ml/datasets/bag+of+words

| Method | Synthetic Data | | News | | IHDP | |
|---|---|---|---|---|---|---|
| | MISE | AMSE | MISE | AMSE | MISE | AMSE |
| original | 0.1414 | 0.0131 | 1.7846 | 0.0100 | 3.2033 | 0.3675 |
| alpha | 0.1413($\Delta$ 0.0% ↑) | 0.0132 ($\Delta$ 0.8% ↑) | 2.6400($\Delta$ 47.9% ↑) | 0.0130 ($\Delta$ 30% ↑) | 3.606($\Delta$ 12.6% ↑) | 0.5115 ($\Delta$ 39.2% ↑) |
| beta | 0.1414($\Delta$ 0.0% ↑) | 0.0137 ($\Delta$ 4.6% ↑) | 2.6547($\Delta$ 48.8% ↑) | 0.0117 ($\Delta$ 17.0% ↑) | 3.7856($\Delta$ 23.5% ↑) | 0.4372 ($\Delta$ 19.0% ↑) |
| gamma | 0.1411 ($\Delta$ 0.2% ↓) | 0.0131 ($\Delta$ 0.0% ↑) | 2.6871 ($\Delta$ 50.5% ↑) | 0.0120 ($\Delta$ 20.0% ↑) | 3.5951 ($\Delta$ 12.2% ↑) | 0.4007 ($\Delta$ 9.0% ↑) |
| re-weighting | 0.1640($\Delta$ 16.0% ↑) | 0.0212 ($\Delta$ 61.8% ↑) | 2.2790($\Delta$ 27.7% ↑) | 0.0105 ($\Delta$ 5.0% ↑) | 3.6804($\Delta$ 14.9% ↑) | 1.1029 ($\Delta$ 200.1% ↑) |

Table 2: Results of ablation study. The top row shows the average MISE and AMSE of DBRNet across 50 runs for the three datasets. Subsequent rows present performance when the respective component is disabled. Numbers in parentheses represent the percentage change in AMSE and MISE relative to the top row results.

## 4.3 RESULTS AND ANALYSIS

To answer **Q1**, we report the mean integrated squared error (MISE) $MISE = \int_{\mathcal{X}}[\int_{\mathcal{T}}(\hat{y}_i(t) - y_i(t))^2 dt]\mathbb{P}(x)dx$, and the mean squared error (AMSE) $AMSE = \int_{\mathcal{T}}[\frac{1}{n}\sum_{i=1}^{n}(\hat{y}_i(t) - y_i(t))]^2\mathbb{P}(t)dt$ to evaluate the performance of the models in estimating the individual level and the population level dose response curve, respectively. Since the integral of $t$ is intractable, we apply all $t$ values existing in the current datasets on each unit to approximate the MISE and AMSE, that is, $M\hat{I}SE = \frac{1}{n}\frac{1}{|\mathcal{T}|}\sum_{i=1}^{n}\sum_{t\in\mathcal{T}}(\hat{y}_i(t) - y_i(t))^2$, $A\hat{M}SE = \frac{1}{|\mathcal{T}|}\sum_{t\in\mathcal{T}}[\frac{1}{n}\sum_{i=1}^{n}(\hat{y}_i(t) - y_i(t))]^2$, where $|\mathcal{T}|$ is the number of different treatment values. To ensure fair and reliable comparisons, we evaluate the performance of all models on 50 repetitions of three different datasets and report the mean and standard deviation of the MISE and AMSE. As presented in Table 1, DBRNet consistently outperforms the majority of baselines across all datasets by a significant margin, achieving satisfactory MISE and the lowest AMSE while demonstrating commendable stable performance. These results demonstrate not only the effectiveness of DBRNet in IDRF estimation but also its ability to adjust for selection bias.

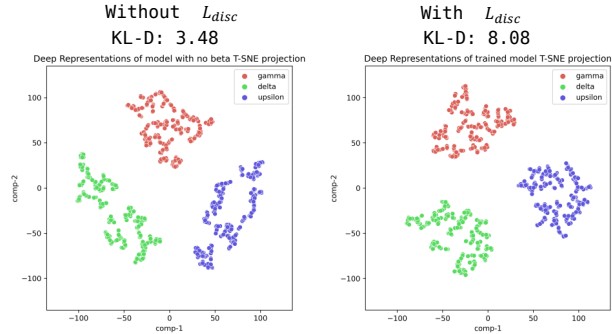

Figure 2: t-SNE plots of the deep representations with/without $L_{disc}$. $KL$-$D$ represents KL-Divergence between three representations.

## 4.4 ABLATION STUDY

To answer **Q2**, we conduct several ablation studies to evaluate the three major components of our model, including the treatment loss $L_T$, discrepancy loss $L_{disc}$, independent loss $L_{ind}$, and the re-weighting function $w(t_i, \Gamma(x_i), \Delta(x_i))$. We demonstrate their roles by setting the corresponding hyperparameter to 0 while keeping the other hyperparameters fixed at the best-tuned values. Especially, to evaluate the re-weighting function, we set the value of re-weighting to 1. As shown in Table 2, all components contribute to the model performance, as evidenced by the significant performance drop in most scenarios when any part is removed. Moreover, we find that the re-weighting function and the discrepancy loss are two critical components of our model due to the significant increase of test error when they are disabled. In other words, adjusting for selection bias and forcing all representations to encode the corresponding information independently are predominant for the model performance. Additionally, to give a visual demonstration of the contribution of the discrepancy loss, we show the t-SNE plots of the model trained with and without $L_{disc}$ on a synthetic dataset in Fig. 2. After incorporating the discrepancy loss, the distinct representations become more separate, leading to a larger KL-divergence.

However, the re-weighting function contributes less to the News dataset than the other dataset due to the distinct data generation process of News. In particular, all features are associated with the treatment assignment and outcome generation, meaning that all features act as confounders. This generation process does not align with the assumptions made in our study and real-world scenarios. Therefore, it becomes challenging for $\Gamma(x_i)$ and $\Upsilon(x_i)$ to learn the correct instrumental and adjustment factors, respectively. Furthermore, inaccurate $\Gamma(x_i)$ and $\Upsilon(x_i)$ can impact the re-weighting function estimation, resulting in a relatively small contribution.

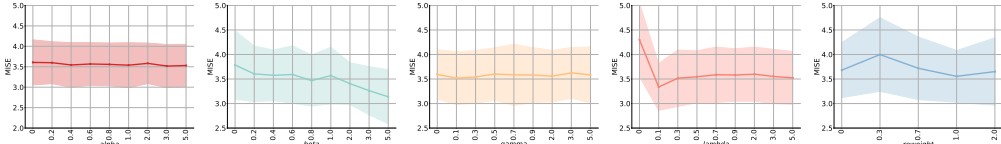

Figure 3: Sensitivity analysis with different values of $\alpha$, $\beta$, $\gamma$, $\lambda$ and different proportions of re-weighting value on 50 repeats of IHDP datasets. The standard deviation band is also plotted.

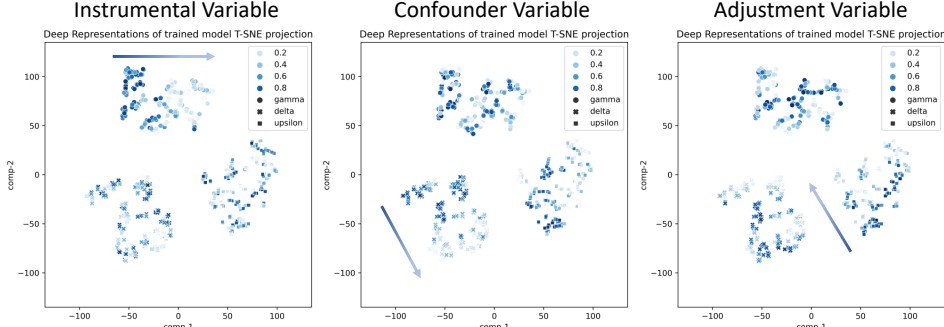

Figure 4: t-SNE plots of the three deep representations with respect to the instrumental variable, confounder variable, and adjustment variable. Point shapes represents the three types of deep representation. Point colors represents the value of the corresponding variable.

**Sensitivity Analysis.** We have also investigated the effects of different values of $\alpha$, $\beta$, $\gamma$, $\lambda$, and different proportions of the re-weighting values on the model performance. The results shown in Fig. 3 suggest that the re-weighting function and the discrepancy loss ($\beta$) have more substantial influence on the model's performance, which consists with the findings from the previous ablation study. Moreover, the results indicate that DBRNet can learn an accurate re-weighting value to improve performance, as evidenced by the fact that the current proportion (1.0) of re-weighting values yields the lowest error in IDRF estimation.

## 4.5 DISENTANGLEMENT PERFORMANCE

To answer **Q3**, we attempt to explore whether the representations capture the corresponding factors by utilizing t-SNE to visualize the three deep representations in a synthetic dataset with respect to different types of variables, as shown in Fig. 4. Since we have knowledge of the data generation process of the synthetic dataset, we know the ground truth of which features are instrumental, confounder, and adjustment variables in the dataset. Hence, for each type of variable, we choose one feature to show the relationship between it and the three deep representations. In Fig. 4, the color shade indicates the value of the corresponding variable, while the shape of the points denotes the type of representation. For example, in the first plot, we aim to verify if the representation of instrumental factors embeds the information of instrumental variables in the covariates. In the representation for instrumental factors (gamma with dot points), the shade of the color varies with the direction of the corresponding arrow, indicating that the instrumental representation encodes some information about this instrumental variable. However, in other representations, such as $\Delta$ and $\Upsilon$ (cross points and squared points, respectively), the color shade does not change regularly, indicating that the confounder and adjustment representations do not encode information about instrumental variables. Therefore, our DBRNet can decently disentangle the three factors.

## 5 CONCLUSION

In this paper, we introduce the Disentangled and Balanced Representation Network (DBRNet), a novel model designed to estimate the individualized dose-response function (IDRF) with high precision while accounting for selection bias through disentangled representations of instrumental, confounder, and adjustment factors. Our experiments on synthetic and semi-synthetic datasets showcase the exceptional disentanglement capabilities of DBRNet and highlight its impressive performance on estimating IDRF, surpassing current state-of-the-art methods.

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

## A    PROOFS

**Lemma 2** *(Importance Sampling) Let $p(x)$ be a probability density for a random variable $X$ defined on $\mathbb{R}^d$, then for any probability density $q(x) \in \mathbb{R}^d$ that satisfies $q(x) > 0$ whenever $f(x)p(x) \neq 0$, we have:*

$$\mathbb{E}_{x \sim p(x)}[f(x)] = \mathbb{E}_{x \sim q(x)}[f(x)\frac{p(x)}{q(x)}].$$

**Proof 1**

$$
\begin{aligned}
\mathbb{E}_{x \sim p(x)}[f(x)] &= \int_{\mathcal{P}} f(x)p(x)dx \\
&= \int_{\mathcal{P}} \frac{f(x)p(x)}{q(x)}q(x)dx \\
&= \mathbb{E}_{x \sim q(x)}[f(x)\frac{p(x)}{q(x)}].
\end{aligned}
\tag{8}
$$

**Theorem 3** *Let $p(x, t')$,$p(x, t)$ denote the counterfactual and factual distribution, respectively. Under the conditions of lemma 2, the expected loss functions for counterfactual and factual outcomes are:*

$$\epsilon_{CF} = \mathbb{E}_{x,t \sim p(x,t')}[l_{\Delta,\Upsilon,g_{\theta(t)}}(x,t)] = \mathbb{E}_{x,t \sim p(x,t)}[l_{\Delta,\Upsilon,g_{\theta(t)}}(x,t)\frac{p(x,t')}{p(x,t)}],$$

$$\epsilon_{F} = \mathbb{E}_{x,t \sim p(x,t)}[l_{\Delta,\Upsilon,g_{\theta(t)}}(x,t)\frac{p(x,t)}{p(x,t)}] = \mathbb{E}_{x,t \sim p(x,t)}[l_{\Delta,\Upsilon,g_{\theta(t)}}(x,t) \cdot 1],$$

*where $t'$ represents all counterfactual treatments of $x$, defined as $t' = \{t'|t' \in \mathcal{T}, t' \neq t\}$. The union of $t$ and $t'$ constitutes the universal set of treatments. Therefore, given the probability density function $f(x,t)$ we have,*

$$\mathbb{P}(x,t) + \mathbb{P}(x,t') = \int_t f(x,t)dt + \int_{\mathcal{T}/\{t\}} f(x,t)dt = \int_{\mathcal{T}} f(x,t)dt = \mathbb{P}(x),\tag{9}$$

The theorem implies that the expected loss function for counterfactual outcomes can be derived by reweighting the factual loss. Hence, we can cooperate the counterfactual loss with factual loss and optimise them together through a designed weight (Hassanpour & Greiner, 2019a);

$$
\begin{aligned}
w =& 1 + \frac{\mathbb{P}(x, t')}{\mathbb{P}(x, t)} \\
=& \frac{\mathbb{P}(x)}{\mathbb{P}(x, t)} \\
=& \frac{1}{\mathbb{P}(t|x)} \\
=& \frac{1}{\mathbb{P}(t|\Gamma(x), \Delta(x))}.
\end{aligned}
\tag{10}
$$

The second equality is by Equation 9, the third equality is by the law of conditional probability, the forth equality is by the Assumption 4 and the independency between adjustment factors and treatment.

**Theorem 4** *(Bias Removal with Weighted Loss) Under the theorem 1 and all assumptions, we have:*

$$
\mathbb{E}[wl_{\Delta, \Upsilon}(x, t)] = \epsilon
$$

**Proof 2**

$$
\begin{aligned}
\mathbb{E}_{x, t \sim p(x, t)}[wl_{\Delta, \Upsilon}(x, t)] =& \mathbb{E}[(1 + \frac{p(x, t')}{p(x, t)})l_{\Delta, \Upsilon}(x, t)] \\
=& \mathbb{E}_{x, t \sim p(x, t)}[l_{\Delta, \Upsilon}(x, t) + l_{\Delta, \Upsilon}(x, t)\frac{p(x, t')}{p(x, t)}] \\
=& \mathbb{E}_{x, t \sim p(x, t)}[l_{\Delta, \Upsilon}(x, t)] + \mathbb{E}_{x, t \sim p(x, t')}[l_{\Delta, \Upsilon}(x, t)] \\
=& \int_{\mathcal{X} \times \mathcal{T}} l_{\Delta, \Upsilon}(x, t)\mathbb{P}(x, t)dxdt + \int_{\mathcal{X} \times \mathcal{T}} l_{\Delta, \Upsilon}(x, t)\mathbb{P}(x, t')dxdt \\
=& \int_{\mathcal{X} \times \mathcal{T}} l_{\Delta, \Upsilon}(x, t)(\mathbb{P}(x, t) + \mathbb{P}(x, t'))dxdt \\
=& \int_{\mathcal{X} \times \mathcal{T}} l_{\Delta, \Upsilon}(x, t)(\mathbb{P}(x))dxdt \\
=& \epsilon
\end{aligned}
\tag{11}
$$

This theorem states that the weighted loss is an unbiased estimation of the IDRF loss, which indicates the re-weighting function in our can precisely eliminate selection bias. Notably, each term in our loss function contributes meaningfully to the theoretical proof. Specifically, some components of the loss function serve to support the assumptions underpinning our proof, as outlined in Assumptions 1-4 in our paper. Their functions are introduced as follows. In Assumption 4, we posit that $X$ should be decomposed into three different factors, where discrepency loss $L_{dis}$ is used to ensure this. $L_{ind}$ enforces us to encode all information of $t_i$ in $\Gamma(x_i)$ and $\Delta(x_i)$ instead of $\Upsilon(x_i)$, thereby facilitating a more precise and accurate estimation of the weight $w$. Moreover, the weight $w$ before factual loss is the key of the proof, hence we use the $L_T$ loss to get the accurate weight.

## B BASELINE METHODS

We compare our model DBRNet with several state-of-the-art methods on continuous treatment effect estimation, including Dragonet, DRNet, VCNet, and TransEE. We did not include non-neural network based models (e.g., GPS (Imbens, 2000)), since they have already been proved to be inferior to neural network based models mentioned above. The details of baselines are as follows.

- **Dragonet**: (Shi et al., 2019) used a three-headed architecture to predict the propensity score and conditional outcome from covariates and treatment information, which discards

| Method | Synthetic Data | IHDP | News |
|---|---|---|---|
| Causal Forest[*][2] | $0.043 \pm 0.0021$ | $0.97 \pm 0.034$ | $0.211 \pm 0.003$ |
| BART[*] | $0.040 \pm 0.0013$ | $\mathbf{0.33 \pm 0.005}$ | $0.066 \pm 0.003$ |
| GPS[*] | $0.028 \pm 0.0016$ | $0.67 \pm 0.025$ | $0.022 \pm 0.001$ |
| Ours | $\mathbf{0.013 \pm 0.0070}$ | $0.37 \pm 0.330$ | $\mathbf{0.010 \pm 0.007}$ |

Table 3: Performance comparision between DBRNet and non-neural network baselines

adjustment information. The model was later improved by(Nie et al., 2021) by using separate heads for treatment in different intervals to adjust for continuous treatments.

- **DRNet**: (Schwab et al., 2020) proposed to divide continuous treatments into several intervals and assign one head to each interval to generate the dose-respond curve. They proposed to make each head more sensitive to treatment by concatenating it into every hidden layer of each head. Following (Nie et al., 2021), DRNet was improved by adding a conditional density estimation head for treatment estimation.

- **Vcnet**: Nie et al. (2021) introduced a varying coefficient structure to allow the prediction head parameters to be functions of continuous treatments. To ensure fair comparisons, in our experiments, we adopt the best hyperparameter settings in the original paper, since the same datasets are used for training and evaluation.

- **TransTEE**: (Zhang et al., 2022) adopted the transformer backbones to estimate treatment effect, where attention mechanisms are used to model treatment interactions, as well as the treatment-covariate which enables further adaptive covariate selection to infer causal effects.

## C  PERFORMANCE COMPARISON BETWEEN DBRNET AND NON-DEEP NEURAL NETWORK MODEL

We conducted our experiments under identical settings with (Nie et al., 2021), including the same data generation process, dataset, and problem setting. Therefore, we compared our methods against non-deep learning approaches such as causal forest, BART, and GPS (reported by (Nie et al., 2021)) on these three datasets. The results from our experiments consistently indicate that our methods exhibit superior, or at the very least, comparable performance in comparison to these alternative approaches.

## D  DETAILS ABOUT THE DIVERGENCE LOSS

The definition of the divergence loss $L_D(\Gamma(x_i), \Delta(x_i))$ is as follows:

$$L_{temp}(\Gamma(x_i), \Delta(x_i)) = \Delta(x_i) \log\left(\frac{\Delta(x_i)}{\Gamma(x_i)}\right) = \Delta(x_i)(\log(\Delta(x_i)) - \log(\Gamma(x_i))) \quad (12)$$

$$L_D(\Gamma(x_i), \Delta(x_i)) = \frac{1}{n} \sum_{n,m} L_{temp}(\Gamma(x_i), \Delta(x_i)) \quad (13)$$

Typically, we do not include the divergence loss between the instrumental representations $\Gamma(x_i)$ and the adjustment representations $\Upsilon(x_i)$. In our model, instrumental $\Gamma(x_i)$ and confounder representations $\Delta(x_i)$ are used to estimate the probability of treatment, while confounder $\Delta(x_i)$ and adjustment representations $\Upsilon(x_i)$ are used to predict the outcome. Hence, instrumental and confounder representations are more similar, as are confounder and adjustment representations, since they are optimized for similar goals. Therefore, we only penalize these. To avoid making our model cumbersome, we choose to not include the divergence loss between instrumental $\Gamma(x_i)$and adjustment representations $\Upsilon(x_i)$. In our work, we specifically use the implementation in PyTorch to compute $L_D$: https://pytorch.org/docs/stable/generated/torch.nn.KLDivLoss.html.

## E  Implementation Details and Hyperparameter Tuning

All the neural network-based methods are trained for 800 epochs with the SGD optimizer (momentum = 0.9). To mitigate the risk of overfitting or underfitting, we apply an early stopping technique. For the three deep representation networks in our model, we implement them as fully connected networks with two hidden layers, and each layer has 50 hidden units using ReLU activation. We also use two-hidden-layer settings (each with 50 hidden units) for the $Y$ prediction network.

The hyperparameters we tuned are as follows: $\alpha, \beta, \gamma \in \{0.1, 0.2, 0.4, 0.6\}$ and the learning rate (lr)$\in \{0.0001, 0.00005, 0.00001\}$. For other hyperparameters, e.g., the number of knots and the degree of B-spline, we follow the setting of (Nie et al., 2021) that is also tuned on the same configurations of datasets. For each dataset, we tune the aforementioned hyperparameters on 20 repetitions. The best hyperparameter settings are as follows:

|       | Synthetic | News   | IHDP    |
|-------|-----------|--------|---------|
| alpha | 0.6       | 0.6    | 0.6     |
| beta  | 0.2       | 0.2    | 0.6     |
| gamma | 0.6       | 0.6    | 0.1     |
| lr    | 0.00001   | 0.0001 | 0.00005 |

## F  Dataset Generation

Our generation process is in line with (Nie et al., 2021). The generation process of the three datasets are as follows:

- **Semi-synthetic Data.** We construct 50 synthetic datasets for training and testing our method and baselines. Each dataset contains 500 training samples and 200 test samples. Furthermore, we simulate another 20 datasets to tune the hyperparameters. In this synthetic dataset all features $x \in \mathbb{R}^6$ follow the i.i.d. distribution,

$$
\begin{aligned}
\tilde{t}|x &= \frac{10\sin(\max(x_1, x_2, x_3)) + \max(x_3, x_4, x_5)^3}{1 + (x_1 + x_5)^2} + \sin(0.5x_3)(1 + \exp(x_4 - 0.5x_3)) \\
&\quad + x_3^2 + 2\sin(x_4) + 2x_5 - 6.5 + \mathcal{N}(0, 0.25) \\
y|x,t &= \cos(2\pi(t - 0.5))(t^2 + \frac{4\max(x_1, x_6)^3}{1 + 2x_3^2}\sin(x_4)) + \mathcal{N}(0, 0.25),
\end{aligned}
\tag{14}
$$

  where $t = (1 + exp(-\tilde{t}))^{-1}$. According to the above equation, $x_2, x_5$ are instrumental variables, $x_1, x_3, x_4$ are confounder variables, and $x_6$ is the adjustment variable.

- **IHDP.** Infant Health and Development Program (**IHDP**), an RCT dataset, originally compiled to estimate the causal effect of binary treatment (home visits of specialists) on future cognitive test scores. The original dataset contains 747 samples with 25 features, in order to estimate continuous causal effect, we generate the treatment and outcome following (Nie et al., 2021). We randomly split the dataset into training set (67%) and testing set (33%).:

$$
\begin{aligned}
\tilde{t}|x &= \frac{2x_1}{(1 + x_2)} + \frac{2\max(x_3, x_5, x_6)}{0.2 + \min(x_3, x_5, x_6)} + 2\tanh(5\frac{\sum_{i \in S_{dis,2}}(x_i - c_2)}{|S_{dis,2}|}) - 4 + \mathcal{N}(0, 0.25), \\
y|x,t &= \frac{\sin(3\pi t)}{1.2 - t}(\tanh(5\frac{\sum_{i \in S_{dis,2}}(x_i - c_1)}{|S_{dis,1}|}) + \frac{\exp(0.2(x_1 - x_6))}{0.5 + 5\min(x_2, x_3, x_5)}) + \mathcal{N}(0, 0.25),
\end{aligned}
\tag{15}
$$

  where $t = (1 + exp(-\tilde{t}))^{-1}$, $S_{con} = \{1, 2, 3, 5, 6\}$ is the index set of continuous features, $S_{dis,1} = \{4, 7, 8, 9, 10, 11, 12, 13, 14, 15\}$, $S_{dis,2} = \{16, 17, 18, 19, 20, 21, 22, 23, 24, 25\}$ and $S_{dis,1} \cup S_{dis,2} = [25] - S_{con}$. Here $c_1 = \mathbb{E}\frac{\sum_{i \in S_{dis,1}}x_i}{|S_{dis,1}|}$, $c_2 = \mathbb{E}\frac{\sum_{i \in S_{dis,2}}x_i}{|S_{dis,2}|}$.

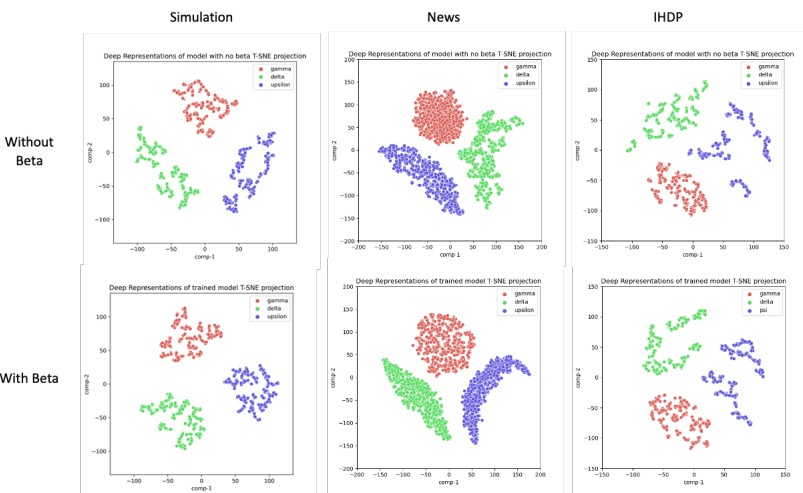

Figure 5: t-SNE plots of the deep represetations with/without $L_{disc}$ across three datasets. For each dataset, we choose one repeat for visualisation due to the space limit.

- **News.** The **News** benchmark is composed of news items from the NY Times corpus, and each item is represented by its word counts. Previous work used it to evaluate the causal effect of devices (binary treatment) on readers' opinions. The original News dataset contains 3000 item samples. Since the dataset is used to estimate the effect of binary treatment, we follow (Nie et al., 2021) to generate continuous treatments as well as the corresponding outcome as follows, we first generate $v'_1, v'_2$ and $v'_3$ from $\mathcal{N}(0, 1)$, and compute $v_i = v'_i / \|v'_i\|_2$. Treatment $t$ is generate from Beta $(2, |\frac{v_3^T x}{2v_2^T x}|)$, where $y$ is generated by:

$$
\begin{aligned}
y'|x, t &= \exp(\frac{v_2^T x}{v_3^T x} - 0.3), \\
y'|x, t &= 2(\max(-2, \min(2, y')) + 20v_1^T x) * (4(t - 0.5)^2 \sin(\frac{\pi}{2}t)) + \mathcal{N}(0, 0.5),
\end{aligned}
\tag{16}
$$

## G   DISCREPANCY LOSS ON OTHER DATASET

In addition, to give a visual demonstration of the contribution of the discrepancy loss, we show the t-SNE plots of models trained with and without $L_{disc}$ on each dataset in Fig. 5. Points belonging to different factors (i.e., $\{\Gamma(x_i), \Delta(x_i), \text{and} \Upsilon(x_i)\}$) are further away under the model trained with the discrepancy loss, indicating that the discrepancy loss indeed helps learn separable representations. Moreover, except for the News dataset, points belonging to the same factor are closer together with $L_{disc}$, suggesting that they extract similar information. However, $\Gamma(x_i)$ in the News dataset becomes more separate, and $\Upsilon(x_i)$ gathers less close compared to other datasets, indicating that it cannot well extract the corresponding information, which is consistent with the explanation of re-weighting above.

## H   AN INSTANTIATION OF CAUSAL GRAPH

In this section, we provide an example of the instantiation of our causal graph. Similar to (Wu et al., 2020): In the context of medical health record, we might collect extensive historical data from each patient, including the patients' features $X$ (e.g., age, gender, living environment, doctor-in-charge), treatment of patients $T$ (taking a particular mediciine or not), and the final outcome $Y$ (cured or not). Among these features, age and gender simultaneously affect the treatment (as a physician would consider these factors when choosing a treatment) and the outcome (since they can also affect the patient's recovery rate); therefore, are confounding factors $\Delta$. In contrast, the doctor-in-charge would influence only the treatment decision, without affecting the outcome, thus being an instrumental

Figure 6: Sensitivity analysis with different values of $\alpha$, $\beta$, $\gamma$, $\lambda$ and different proportions of re-weighting value on 50 repeats of IHDP datasets. The standard deviation band is also plotted.

factor $\Gamma$. Environment, which only affects the outcome but not the treatment, falls into the category of adjustment factors $\Upsilon$.

# I    SENSITIVITY ANALYSIS BASED ON AMSE

In this section, we present a sensitivity analysis based on AMSE, as shown in Figure 6. The trend observed is similar to that of the MISE, which is presented in the main text. This analysis demonstrates that the re-weighting function and the discrepancy loss $\beta$ have a more substantial influence on the model's performance. These findings are consistent with those from our previous ablation study.

