# OpenReview forum: "DBRNet: Advancing Individual-Level Continuous Treatment Estimation through Disentangled and Balanced Representation"
_ICLR.cc/2024/Conference — Submitted to ICLR 2024_

### Official Review · Reviewer_s6Cp · 2023-10-14

**Soundness:** 3 good
**Presentation:** 3 good
**Contribution:** 3 good
**Rating:** 8
**Confidence:** 4

**Summary:**

In this paper, authors have studied the continous treatment effect estimation problem and proposed a novel disentangled and balanced representation network (DBRNet) for estimating the individualized dose-response function (IDRF) that precisely balances the confounders. DBRNet learns disentangled representations using a loss term based on Kullback-Leibler (KL) divergence between latent factors and imbalance loss term. Moreover, selection bias is addressed by using inverse propensity score based weighting to the factual loss term. They used synthetic and two semi-synthetic datasets to evaluate the model against the existing techniques.

**Strengths:**

- This is a novel method for continuous treatment setting which combines two ideas of disentanglement and weighting, and I find that as sufficient novelty. In fact, they extended Hassanpour and Greiner [2019b] paper from discrete to continuous setting. Authors also claim this to be first work that precisely balances the confounders for the continuous treatments.
- Overall, clearly written and well-organized.
It was easy to follow. Introduction clearly presents the problem and motivations for the work, and related work puts the work in context, and presents the literature gap. Similarly, methdology is also presented with sufficient details.
- Sufficient experiments with detailed ablation study to prove the effficacy of the method as well as different components of the idea. Paper clearly formulates the research questions and provide empirical evidence for each.
- Paper provides theoretical substantiation that the devised re-weighting function is able to precisely adjust for selection bias.

**Weaknesses:**

**NOTE:** I reviewed this work earlier as part of NeurIPS. I can notice the changes authors have done to the paper.

However, it is not clear if authors have addressed my concern about under-training/over-training due to training all algorithms for fixed 800 epochs (and not using something like early stopping).

**Questions:**

Can you confirm if you have addressed the under-training/over-training issued caused by training for fixed 800 epochs?

---

> ### Author Response · Authors · 2023-11-18
> **Rebuttal by Authors**
>
> Thank you for your valuable insights and the recognition of the improvements in our work. Your feedback as an ICLR reviewer, with experience from NeurIPS, is greatly appreciated.
>
> Regarding your concern about the use of 800 epochs for every model, we agree with your point. To address this, we have implemented an early stopping technique across all models to prevent overfitting and underfitting in this ICLR version. This modification is detailed in our code. Furthermore, we have specifically highlighted this change in blue on page 13 in the Implementation Details section of our revised manuscript.
>
> We trust that these adjustments adequately address your concerns. We are dedicated to integrating your constructive feedback into the revised version of our paper. Once again, we extend our gratitude for your insightful comments.

---

> > ### Comment · Reviewer_s6Cp · 2023-11-18
> >
> > Dear authors, thanks for the clarifications.
> >
> > As my concerns are addressed so I have reaised my score. All the best!

---

### Official Review · Reviewer_8hoc · 2023-10-30

**Soundness:** 2 fair
**Presentation:** 3 good
**Contribution:** 2 fair
**Rating:** 3
**Confidence:** 4

**Summary:**

The paper proposes DBRNet, a neural method leveraging disentangled representations representing different underlying latent variables (adjustment factors, instrumental variables, confounding variables)  to estimate the individualized dose-response curve. For this, they adapt the method of [1] to the setting of continuous treatments leveraging parts of the work from [2]. The paper further introduces an additional discrepancy loss term. Finally, the method is evaluated on existing synthetic and semi-synthetic datasets used by previous works.

**Strengths:**

1. IDRF estimation is an important topic, especially in domains such as medicine.
2. The main contribution of the paper (disentangling latent factors for continuous treatment setting) is well motivated and easy to understand.

**Weaknesses:**

In general, there are multiple points in the evaluation of the paper which seem problematic or unclear to me:
1. The paper is mainly motivated by learning the **individualized** dose response function. This is a different task than learning the **average** dose response function. Hence, the main metric for evaluation should be the MISE and not the AMSE (even though it is a nice add-on but not necessary). Thus, the MISE should also be the displayed metric e.g., in Fig. 3 and Table 3, and the main interpretation of the results should focus on the MISE.
2. My understanding is that the MISE is not properly implemented (see also code). The paper approximates the integral over the treatments in the MISE by averaging over the existing treatments in the datasets $t \in \mathcal{T}$. However, since the datasets are biased with respect to the treatment assignment, this may yield biased estimates of the MISE. Instead one could use Romberg integration (as in [3]) are just simply use at least equally spaced points in $\mathcal{T}$
3. The sensitivity analysis in Fig. 3 states in the description: “ Sensitivity analysis with different values of $\alpha, \beta$”, but the axis of the plots are $\beta$ value and $\gamma$ value which is confusing. Also, the plots for $\beta$ and $\gamma$ do not show clear performance gains outside of the standard deviation bounds. This may indicate that the (quite substantial) performance gains reported in Table 2 are not significant because of the high variance of the method, or may be quite prone to hyperparameter tuning and model selection. Clarification here would be much appreciated.
4. The paper argues that existing methods using selection bias adjustment “fail to accurately adjust for selection bias, as they do not make any adjustments to eliminate the bias, or resort to a simple and sophisticated approach to balance the entire representation.” While there is work on binary treatments showing this, it is important to benchmark against such methods in a continuous setting to support this argument, e.g., using the balancing applied in [4] as a baseline.
5. The theory is primarily around re-weighting and not balancing.

Minor: There are a many spacing issues with missing spaces around references.

[1] Hassanpour, Negar, and Russell Greiner. "Learning disentangled representations for counterfactual regression." International Conference on Learning Representations. 2019.
[2] Nie, Lizhen, Mao Ye, and Dan Nicolae. "VCNet and Functional Targeted Regularization For Learning Causal Effects of Continuous Treatments." International Conference on Learning Representations. 2020.
[3] Schwab, Patrick, et al. "Learning counterfactual representations for estimating individual dose-response curves." Proceedings of the AAAI Conference on Artificial Intelligence. Vol. 34. No. 04. 2020.
[4] Bellot, Alexis, Anish Dhir, and Giulia Prando. "Generalization bounds and algorithms for estimating conditional average treatment effect of dosage." arXiv preprint arXiv:2205.14692 (2022).

**Questions:**

1. I assume the “_TR” in the baseline methods refers to targeted regularization. However, TR is designed for ATE / ADRF estimation. Motivation about why these baselines are included (and still oftentimes outperform methods without TR even in MISE) would be appreciated.
2. Why is alpha not included in the ablation studies? Its contribution could also be interesting, especially since only a very simplified density estimator is used and it could show if this still leads to performance gains.
3. The discrepancy loss in Eq (5.) leverages KL divergence between representations. However, as I understand, the learned representations are not probabilistic so how is the KL divergence approximated which is defined as a measure between two probability distributions? Also, if there are some implicit assumptions here which are necessary to ensure statistical independence between the representations they should be stated.

---

> ### Author Response · Authors · 2023-11-18
> **Rebuttal by Authors**
>
> 1.**Include more MISE results**
>
> We are grateful for your insightful suggestion regarding the emphasis on MISE. Initially, we had prioritized AMSE as the primary metric due to its popularity. However, after carefully considering your valuable suggestion, we agree that focusing on MISE is indeed more appropriate for our paper. To reflect this, we have thoughtfully reordered the tables to give precedence to MISE over AMSE. Additionally, we have revised the displayed metric in Fig. 3 according to your suggestion. We have also refined the description of our results to highlight the MISE findings more prominently. All these modifications have been highlighted in blue for ease of review.
>
> 2.**Implementation about the MISE**
>
> Thank you for your reminder. We have updated the code to include the implementation of the MISE. Additionally, we would like to clarify that our test dataset (data used to compute MISE and AMSE) is not biased. To approximate the AMSE and MISE, we apply all $t$ values existing in the current dataset to **each unit**. Consequently, one unit is tested on all available $t$ values, ensuring that the treatment assignment is not influenced by selection bias. Given this unbiased dataset, we believe that using all real $t$ values in the dataset is preferable to equally sampled points in $\mathcal{T}$. This approach better represents the actual $t$ distribution, as opposed to generating equal-interval $t$ values.
>
> We have carefully reviewed the paper you suggested. In their approach (Equation 1 on page 3), they also proposed to use all $t$ values in the dataset. However, in their implementation, they employ Romberg integration, as you mentioned. We conjecture this might be related to their method, which involves dividing the treatment equally into several strata and assigning a specific prediction head for each. Romberg integration, in such case, is a more suitable metric for their method as it ensures accurate predictions for each head. Nevertheless, in our paper, we respectfully argue that using the actual $t$ values from the dataset may be a better choice for our methodology.
>
> 3.**Clarification on sensitivity analysis**
>
> We appreciate your detailed review and have corrected the typo. **Furthermore, we have expanded the test range of hyperparameters and included all hyperparameters in the sensitivity analysis. The new results can be found in the revised paper.** The figure shows that the re-weighting function and the discrepancy loss ($\beta$) have more substantial influence on the model's performance, which consists with the findings from the previous ablation study. The performance improves notably when $β$ is larger and the re-weighting factor is set to 1.
>
> 4.**Add more ablation studies for alpha**
>
> We agree with you that including all hyperparameters would likely make our analysis more robust. Our initial decision to exclude certain hyperparameters was based on previous studies indicating their minimal impact on results [3], a finding consistent with our preliminary findings. However, as you suggested, exploring this further could be interesting. Therefore, **we we have added an ablation study for $α$ and a sensitivity anaylsis for $α$ and $λ$ based on both MISE and AMSE in our revised paper**. Due to the space limit we only show the figure based on MISE as you suggested, and we put the result of AMSE in Appendix I. The results confirm our alignment with prior findings: while including this hyperparameter ($\alpha$) does improve performance, the improvement is slight.
>
> 5.**Baseline Method**
>
> Thank you for your suggestion. We would love to add this baseline; however, the code is not open-sourced, and despite our efforts to contact the author, we have not received a response. Should the code be released, we will certainly do our best to include it as a baseline.
>
> 6.**Clarification on the proof**
>
> We agree with your observation. In our paper, we utilize re-weighting to adjust for selection bias, and the provided proof aims to offer theoretical support for this approach. Throughout the paper, we clarify that the proof pertains to the debiasing ability of the re-weighting method rather than balancing. We would like to understand more about your reference to 'balancing' to address any specific concerns you might have.
>
>
> 7.**Missing space Problem**
>
> We appreciate your detailed review and **have already corrected the missing space.**

---

> ### Author Response · Authors · 2023-11-18
> **Rebuttal by Authors**
>
> 8.**Clarification on TR**
>
> As you rightly pointed out, TR represents targeted regularization.
> As you mentioned, TR is designed for ATE / ADRF estimation. Furthermore, to obtain ATE and ADRF, the papers [2-3] (which proposed TR) actually first calculate the individual treatment effect and then average them to derive the ATE and ADRF. We conjecture that thoes approaches may also lead to better individual treatment effect estimation. Otherwise, the papers [2-3] might not achieve improved ATE/ADRF results through averaging if individual treatment effect were inaccurately estimated. Therefore, we believe it would be interesting to compare these with our methods. Moreover, including more baseline methods also makes the results more convincing.
>
> 9.**Clarification on Discrepancy Loss**
>
> And we agree with you that the KL divergence is used to measure the discrepancy between two distributions of representations. Typically, in our paper, we use the KL divergence loss, which slightly differs from the original KL divergence. Here is the detailed PyTorch implementation of the KL loss that we use: https://pytorch.org/docs/stable/generated/torch.nn.KLDivLoss.html KL Divergence Loss. **Thanks for your question. In response, we have added a section in Appendix D, highlighted in blue, to further clarify KL divergence. Additionally, we have included a sentence in the methods section to guide readers to this new Appendix section.**
>
> We hope that our response addresses your concerns adequately, and we are committed to incorporating your valuable feedback into the revised version of our paper. Thank you once again for your insightful comments.
>
> [1] Schwab P, Linhardt L, Bauer S, et al. Learning counterfactual representations for estimating individual dose-response curves[C]//Proceedings of the AAAI Conference on Artificial Intelligence. 2020, 34(04): 5612-5619.
>
> [2] Shi C, Blei D, Veitch V. Adapting neural networks for the estimation of treatment effects[J]. Advances in neural information processing systems, 2019, 32.
>
> [3] Nie L, Ye M, Nicolae D. VCNet and Functional Targeted Regularization For Learning Causal Effects of Continuous Treatments[C]//International Conference on Learning Representations. 2020.

---

> > ### Comment · Reviewer_8hoc · 2023-11-20
> >
> > I would like to thank the authors for their answers. I appreciate the updates regarding the evaluation metrics and the additional sensitivity analyses. However, multiple points remain unaddressed and do not seem sound to me. In particular, among other points, simply applying the PyTorch implementation of the KLL divergence loss on top of the representations lacks theoretical foundation for guaranteeing independence, as no probabilistic distributions over the latent factors are learned. Here, I share the concerns of **reviewer cz1b**. Hence, I do not feel comfortable updating my score.

---

> > > ### Author Response · Authors · 2023-11-20
> > > **Further Clarification**
> > >
> > > Dear Reviewer,
> > >
> > > Thank you for acknowledging our updates. We would like to further clarify the implementation of the independent loss. In our paper, the intent of the independent loss is to **make the three representations more separate from each other**. We were inspired by the KL divergence and the KL divergence loss in PyTorch to design this loss. After carefully considering your comments and those of Reviewer cz1b, we agree that referring to it as 'KL divergence' is not appropriate. As you mentioned, KL divergence is typically used to calculate the discrepancy between probability distributions, which is indeed different from ours. Therefore, we have decided to rename this loss as 'divergence loss' $L_D$ and include a full definition of this loss in our revised paper to avoid confusion (the modifications are highlighted in blue). However, we kindly argue that **our loss is still effective for separating the representations**.  for example, $L_D(\Gamma(x_i), \Delta(x_i))$ is calculated as follows :
> > >
> > > $L_{temp}(\Gamma(x_i), \Delta(x_i)) = \Delta(x_i)\log\left(\frac{\Delta(x_i)}{\Gamma(x_i)}\right) = \Delta(x_i)(\log(\Delta(x_i)) - \log(\Gamma(x_i)))$
> > >
> > > $L_D(\Gamma(x_i), \Delta(x_i)) = \frac{1}{n}\sum_{n,m} L_{temp}(\Gamma(x_i), \Delta(x_i)),$
> > >
> > > where $n$ and $m$ represent the batch size and the latent factor dimension, respectively.
> > >
> > > Using this formula, if the representations $\Gamma(x_i)$ and $\Delta(x_i)$ are identical, the result will be 0. Conversely, if $\Gamma(x_i)$ and $\Delta(x_i)$ are distinctly different, the value will be larger. Therefore, we utilize the reverse of this loss to ensure that the representations are **separate** from each other, thereby guaranteeing that each component learns only the information relevant to its specific functionality. To demonstrate the effectiveness of this loss in our model, we present t-SNE plots of the model trained with and without $L_{disc}$ on a synthetic dataset in Fig.2. The incorporation of the discrepancy loss results in more distinct representations and a higher KL-divergence score, validating our loss approach.
> > >
> > > We would like to inquire if there are any points that remain unaddressed in your view, as we believe we have responded to **all nine questions**. Your advice and suggestions are invaluable to us.

---

### Official Review · Reviewer_TzGH · 2023-10-31

**Soundness:** 2 fair
**Presentation:** 2 fair
**Contribution:** 2 fair
**Rating:** 5
**Confidence:** 3

**Summary:**

This paper aims at estimating individual-level continuous treatment effects, a crucial aspect in fields like personalized healthcare and customized marketing. It extends existing work and tries to address the limitations of existing methods that focus on discrete treatments via important reweighting. Extensive experiments on synthetic and semi-synthetic datasets demonstrate the effectiveness of the proposed method.

**Strengths:**

1. Attempting to Tackle a Challenging Problem: The paper targets a crucial area in the field of causal inference, specifically the estimation of individual-level continuous treatment effects.
2. Theoretical Analysis: The authors provide theoretical analysis to support the effectiveness of the debiasing process employed in DBRNet.

**Weaknesses:**

1. **Proof of Major Results (Equation 9)**:
   - In the proof citing the law of total probability, it seems to assume binary treatment. This appears inconsistent with the claim that the method is capable of handling continuous treatments.

2. **Self-Containment and Clarity**:
   - The process of identifying and estimating the counterfactual distribution $p(x, t')$ from observed data, especially when $t'$ is a continuous variable, is not sufficiently detailed. More explicit steps or examples would greatly benefit readers in understanding this crucial aspect of the proposed methodology.
   - The authors use the independent loss for disentanglement. It's important to note that this doesn't necessarily imply full identifiability of the real underlying causal factors. Given the known challenges in achieving full identifiability in nonlinear independent component analysis (up to a component linear transformation), it would be helpful to discuss this limitation and its effect in the paper. A more rigorous proof or argument regarding the identifiability and independence of representations in the model would be beneficial.

**Questions:**

- Could the authors elaborate on how the law of total probability is applied in Equation 9, particularly in continuous treatments, to align with the method's main contribution?
- How exactly is the counterfactual distribution $p(x, t')$ identified and estimated from observed data when dealing with continuous treatments? Could the authors provide more detailed explanations or examples in this regard?
- Given the challenges in achieving full identifiability in nonlinear independent component analysis, could you discuss how the authors' approach counters this issue? What is the influence of not achieving full identifiability with the proposed methods?
- Could the authors provide rigorous proof or detailed argument to support the claim of independence and identifiability in the proposed model's representations?
- What does the $\gamma$ symbolize in Figure 3? What does TR mean in Table 1?
- Could the authors further explain the specific techniques or methods leveraged in the paper to handle continuous variables compared to Hassanpour and Greiner [2019b]?

---

> ### Author Response · Authors · 2023-11-18
> **Rebuttal by Authors**
>
> To Reviewer TzGH:
>
> 1.**Clarification on the law of total probability**
>
> Thank you for pointing this out. We realized that the equation and term were confusing, so we have carefully revised them. A new paragraph has been added to Appendix A, which includes our revised equation for continuous settings, highlighted in blue. We have also included additional explanations in the proof section, similarly highlighted. Typically,  $t'$ represents all counterfactual treatments of $x$, defined as $t'= \\{t'| t'\in \mathcal{T}, t'\neq t \\}$. The union of $t$ and $t'$ constitutes the universal set of treatments. Therefore, given the probability density function $f(x,t)$ we have, $ \mathbb{P}(x,t)+\mathbb{P}(x,t')=\int_{t}f(x,t)dt+\int_{\mathcal{T} / \{t\}}f(x,t)dt
> =\int_{\mathcal{T}}f(x,t)dt
> =\mathbb{P}(x)$
>
> 2.**Estimation of counterfactual distribution, and the technique to deal with the continuous treatment estimation**
>
> In our paper, we do not estimate the counterfactual distribution $p(x, t')$; instead, our goal is to estimate the IDRF as shown in Equation 1 $\mu(t, x) = \mathbb{E}[Y(T=t)|X=x, T=t]$. This involves estimating the corresponding $y$ for any given $x$ and $t$. We acknowledge, as you rightly pointed out, that addressing this continuous treatment estimation is a crucial aspect of our work. Typically, we adopt the varying coefficient network [2] (as introduced in Section 3.2 in factual loss) to estimate treatment effects under continuous settings. **Thank you for pointing out the confusion of the function of the varying coefficient network. We have added a sentence highlighted in blue in the methods section, to elucidate that.**
>
> In binary settings, as demonstrated in the paper by Hassanpour and Greiner [2019b], the common approach is to learn a deep representation of $x$ and then construct a neural network for each treatment to estimate $\mu(t, x)$. However, this method cannot be extended to continuous settings due to the infinite number of treatments, making it impractical to build a neural network for each value of $t$. Therefore, we utilize the varying coefficient network [2] (introduced in factual loss in section 3.2), which enables us to estimate treatment effects under continuous settings. Specifically, the varying coefficient structure uses a function $g_{\theta(t)}(\Delta(x_i), \Upsilon(x_i))$ with varying parameters $\theta(t)$ rather than fixed parameters to predict outcomes $y$ given the treatment $t$ and the corresponding deep representation of covariates $\Delta(x_i), \Upsilon(x_i)$. In this model, the parameters are determined by $t$, allowing us to incorporate the continuous $t$ in outcome estimation and estimate $\mu(t, x)$ without the need to construct a neural network for each value of $t$.
>
> Particularly, we employ a B-spline of degree $p$ with $q$ knots, resulting in $k=p+q+1$ basis functions. Let $\textbf{B} =[b_1, b_2, ..., b_k] \in \mathbb{R}^{n\times k}$ denote the spline basis for the treatment $T \in \mathbb{R}^{n\times 1}$. For a single-layer feedforward network with $p$ inputs and $q$ outputs, the function is given by $f_{\theta(t)} = \sum_{i=1}^{k} (b_i \cdot (\textbf{X}\textbf{W}))$, where $\textbf{W} \in \mathbb{R}^{p\times q\times k}$ is the optimizable weight matrix.
>
> 3.**Identifiability of the latent factors**
>
> We respectfully argue that our method differs from nonlinear independent component analysis, and our latent factors are fully identifiable through supervised learning. We acknowledge your point that achieving full identifiability in independent component analysis is challenging, as it is inherently unsupervised. While it can separate the latent factors, identifying the semantic meaning of each latent factor is difficult. In contrast, our method leverages **supervised learning** to ensure that each deep representation learns its corresponding information effectively.
>
> In our paper, we describe three types of deep representations:
> 1. Instrumental factors $\Gamma(x)$, which are associated with the treatment but not with the outcome except through the treatment.
> 2. Confounder factors $\Delta(x)$, which are associated with both the treatment and the outcome.
> 3. Adjustment factors $\Upsilon(x)$, which are predictive of the outcome but not associated with the treatment.
>
> We use instrumental and confounder factors to predict treatment, and confounder and adjustment factors to predict the outcome. The ground truth of the treatment and the outcome, along with the loss function (Equation 2), are used to optimize the learned representations (supervised learning). This approach ensures that different representations only encode relevant information for their respective roles, thereby achieving identifiability. In addition, we attempt to explore whether the representations capture the corresponding factors by utilizing t-SNE to visualize the three deep representations. The results in section 4.5 show that our method has the ability to identify the corresponding factors.

---

> ### Author Response · Authors · 2023-11-18
> **Rebuttal by Authors**
>
> 4.**Terms' meaning**
>
> Thank you for pointing out this issue, which was indeed a typo in the caption of Figure 3. **We have now corrected it**. The symbol $γ$ represents the weight before the independent loss $L_{ind}$ (as detailed in Equation 2 on page 4) and is one of the hyperparameters. Figure 3 is intended to illustrate the sensitivity of these hyperparameters. TR refers to targeted regularization, a technique that improve the accuracy of treatment estimation [1,2]. We appreciate your observation and **have added the full name about this term to enhance clarity**.
>
> 5.**More details about the methods to handle continuous variables**
>
> Please refer to our response in answer 2 for the clarification. We hope this addresses your question. Should you have any further queries or require additional information, please do not hesitate to let us know.
>
>
> We hope that our response addresses your concerns adequately, and we are committed to incorporating your valuable feedback into the revised version of our paper. Thank you once again for your insightful comments.
>
> [1] Shi C, Blei D, Veitch V. Adapting neural networks for the estimation of treatment effects[J]. Advances in neural information processing systems, 2019, 32.
>
> [2] Nie L, Ye M, Nicolae D. VCNet and Functional Targeted Regularization For Learning Causal Effects of Continuous Treatments[C]//International Conference on Learning Representations. 2020.

---

> ### Author Response · Authors · 2023-11-21
> **Further Discussion**
>
> Dear Reviewer,
>
> Thanks for your assessment and your encouragement of our work. As the deadline for reviewer-author discussion (11/22/23) approaches, we would be happy to take this opportunity to make more discussions about any of our questions or concerns. If our response has addressed your concerns, we would be grateful if you could re-evaluate our paper based on our feedback and new results.
>
> Sincerely,
>
> Authors

---

> ### Comment · Reviewer_TzGH · 2023-11-23
> **Concerns about Theorem 1**
>
> Dear authors,
>
> Could you provide rigorous definition of each term and detailed explanation about Theorem 1. It seems that the assumption of P(x, t') for making the reweighting holds can not be defined by following the definition of t' in your proof.
>  t' represents all counterfactual treatments does not imply P(x, t') is a counterfactual distribution. The counterfactual distribution should be P'(X,T), where X and T are random variables.
>
> In this case how can we know the factual treatment t is not allowed in the counterfactual world?

---

> > ### Author Response · Authors · 2023-11-23
> > **Clarifications on Theorem 1**
> >
> > Dear Reviewer,
> >
> > We would like to further clarify the definition of Theorem 1. Typically, $ p(x, t) $ represents the joint distribution of a unit $x$ and its corresponding factual treatment, denoted as $p(x, t)$ (factual distribution for $x$). Conversely, $p(x, t')$ (counterfactual distribution for $x$) denotes the joint distribution of unit $ x $ and any corresponding counterfactual treatment, where $ t'$ can not be equal to $t$ ensuring that the factual treatment $t $ is not present in the counterfactual world, as represented by $ t' = \\{t' | t' \in \mathcal{T}, t' \neq t \\} $ . Therefore, we have:
> >
> > $ \mathbb{P}(x, t) + \mathbb{P}(x, t') = \int_{t} f(x, t) \, dt + \int_{\mathcal{T} \setminus \{t\}} f(x, t) \, dt = \int_{\mathcal{T}} f(x, t) \, dt = \mathbb{P}(x) $
> >
> > We acknowledge there was confusion regarding the definition in Theorem 1, and we have made clarifications in the latest version of our manuscript.
> >
> > We hope this addresses your concerns.
> >
> > Best regards,
> >
> > The Authors

---

> ### Comment · Reviewer_TzGH · 2023-11-23
> **Clarifications on Theorem 1 (continue)**
>
> - From your definition, it seems $P(X, T')$ is not a distribution anymore as $\int_{support(P(X, T'))}P(x, t')dxdt'= 1-\int_{x}P(x,t)dx<1$, where X and T' are random variables of x and t'.
> - Additionally, how do you empirically calculate error CF in Theorem 1 step-by-step. It seems that we need to have a sample from the counterfactual distribution.

---

> > ### Author Response · Authors · 2023-11-23
> > **Clarifications on Theorem 1**
> >
> > Dear Reviewer,
> >
> > We understand your point and agree that the term "distribution" may lead to confusion. Upon carefully reviewing the lemma and theorem, we think  that the use of importance sampling theory for reweighting (Lemma 1) does not strictly require a distribution. As long as $ p(x)$ and $ q(x) $ are within the real numbers $ R$ and represent probability density (not probability density functions), then the reweighting approach is valid.
> >
> > Best regards,
> >
> > The Authors

---

> > > ### Comment · Reviewer_TzGH · 2023-11-23
> > > **Any support**
> > >
> > > Could you find any support for this claim? If it is not a distribution, it is not any expectation anyone.
> > >
> > > I have other questions are that if the "counterfactual distribution" is the same is the observed distribution except not having t'.
> > > - Why do we need to tell our story from perspective of counterfactual?
> > > - why do we need reweighting?

---

> > > > ### Author Response · Authors · 2023-11-23
> > > > **Clarifications on Theorem 1**
> > > >
> > > > Dear Reviewers,
> > > >
> > > > Equation 2 in [1] and Equation (6.2) from [this source](https://www.math.arizona.edu/~tgk/mc/book_chap6.pdf) provide support for our approach.
> > > >
> > > > The rationale for considering counterfactual scenarios is that we aim to estimate the Individual Dose-Response Function (IDRF), denoted as $ \mu(t,x)= \mathbb{E}[Y(T=t)|X=x, T=t]$. For any given unit, we seek to estimate the treatment effect for any $ t $; however, for a single unit, we can only observe one \$t $, with all others being unobserved (counterfactual). Counterfactual scenarios are crucial, particularly in medical applications, where we may wish to determine the outcome for a patient if they were to take any dosage of medication, to find the optimal dosage. Clearly, it is impractical to have the patient actually take every possible dosage. Hence, counterfactual analysis is essential.
> > > >
> > > > Reweighting aims to eliminate selection bias, as confounder factors may simultaneously affect the treatment and outcome, leading to an unbalanced distribution of certain features across different groups. For example, in binary settings, if the treatment group is composed of 90% elderly people and 10% young people, while the control group is mostly young, age becomes a confounding factor. This imbalance between the treatment and control groups could bias the treatment estimation since age is not controlled. Therefore, in our paper, we employ reweighting to adjust for selection bias, and our proofs further elucidate this point.
> > > >
> > > > We hope this addresses your concerns. We apologize that we cannot take further questions at this moment due to the time difference around the world—it is already 4 **am** here.  Happy Thanksgiving!
> > > >
> > > > Best regards,
> > > > The Authors
> > > >
> > > > [1]Tokdar S T, Kass R E. Importance sampling: a review[J]. Wiley Interdisciplinary Reviews: Computational Statistics, 2010, 2(1): 54-60.

---

### Official Review · Reviewer_cz1b · 2023-11-06

**Soundness:** 2 fair
**Presentation:** 2 fair
**Contribution:** 2 fair
**Rating:** 5
**Confidence:** 3

**Summary:**

This paper has proposed DBRNet, which learns disentangled and balanced representations for continuous treatment effect estimation at the individual level. DBRNet is the first model that could adjust for selection bias in continuous treatment settings.

**Strengths:**

1. This paper is well-motivated, "how to adjust for selection bias in continuous treatment settings" is an important research problem that existing works haven't solved.

2. This paper is well-organized, it is easy to get the main ideas for readers.

**Weaknesses:**

1. The proposed method relies heavily on the causal graph shown in Figure 1(a). However, the authors haven't provided a sufficient explanation for the soundness of this causal graph. The authors should at least provide several specific examples (such as Johansson et al., 2016) about what $\Gamma, \Delta, \Upsilon$ respectively represents.

2. In Section 3.2, the authors formulate the loss function of their proposed DBRNet, it is too sophisticated and contains too many hyper-parameters. As the authors claimed, the discrepancy loss encourages $\Gamma, \Delta, \Upsilon$ to be independent of each other, and the independent loss encourages $\Upsilon, T$ to be independent of each other. However, based on the causal graph shown in Figure 1, why not DBRNet contain another term to encourage $\Gamma, Y$ to be independent of each other given $T, \Delta$? Besides, there are some questions about the soundness of the discrepancy loss and independent loss, please see Questions.

3. It seems that the theoretical results only focus on factual loss, so there exists a remarkable gap between the sophisticated loss function and the theoretical analysis.

4. I'm happy that the authors have performed sensitivity analysis w.r.t. $\alpha, \beta$, so why not perform that w.r.t. $\gamma, \lambda$?

**Questions:**

Discrepancy loss minimizes KL divergence between $\Gamma(x_i)$ and $\Delta(x_i)$ and the authors claimed that if $\Gamma(x_i)$ and $\Delta(x_i)$ are independent of each other, the KL divergence is 1. First, considering $x_i$ is a realization of the random variable X (rather than a random variable), what does "$\Gamma(x_i)$ and $\Delta(x_i)$ are independent of each other" mean? Second, in my experience, KL Divergence can only measure the similarity between two distributions but not the independence of two random variables. The similar question also exists for independence loss. If I'm wrong, please provide some references.

---

> ### Author Response · Authors · 2023-11-18
> **Rebuttal by Authors**
>
> To Reviewer cz1b
>
> 1.**Explanation and soundness of the causal graph**
>
> We agree with you that utilizing a concrete example can help readers better understand the causal graph and enhance its soundness, and thus we **have added a section in Appendix H (An instantiation of causal graph) and the detailed motivation and intuition for the disentanglement in the method section.(both are highlighted in blue)**.
>
> The motivation and intuition behind the disentanglement in our study are rooted in causal inference theory, which typically classifies covariates (input features) into three categories: instrumental variables (predictive only of treatment), confounder variables (influencing both treatment and outcome), and adjustment variables (predictive solely of outcome). Furthermore, disentangling $X$ into these three factors is a commonly used approach in causal inference research, as evidenced in [1, 2, 3, 4, 5], allowing for the precise extraction of relevant information.
>
> We presented a detailed example in a medical scenario, as proposed in [1]: We might collect extensive historical data from each patient, including the patients' features $𝑋$ (e.g., age, gender, living environment, doctor-in-charge), treatment of patients $𝑇$ (taking a particular medicine or not), and the final outcome $𝑌$ (cured or not). Among these features, age and gender simultaneously affect the treatment (as a doctor would consider these factors when choosing a treatment) and the outcome (since they can also impact the patient’s recovery rate), hence they are confounding factors $𝐶$. In contrast, the doctor-in-charge would influence only the treatment decision, without affecting the outcome, thus being an instrumental factor $𝐼$. Environment, which only affects the outcome but not the treatment, falls into the category of adjustment factors $𝐴$.
>
> 2.**Clarification on KL divergence**
>
> Thank you for prompting us to clarify our approach to KL divergence. The aim of our causal graph is to demonstrate that covariates can be decomposed into three factors: $Γ(x_i), Δ(x_i), Υ(x_i)$, where $Γ(x_i)$ and $Δ(x_i)$ should be used to estimate $t_i$, while $Δ(x_i)$ and $Υ(x_i)$ should be used to estimate $Y$. Hence, our model structure is already fulfilled this causal graph, and the rest of loss is mainly aim to adjust selection bias.
>
> The disentanglement loss $L_{dis}$ is designed to ensure that these factors are more separated and encode only the relevant information. The purpose of adding an independent loss $L_{ind}$ is to eliminate the selection bias caused by biased treatment assignment. Since we aim to prevent the distribution of the adjustment factor $Υ(x_i)$ from varying across different treatments, so we introduce this term (make the adjustment factor to be independent of $t_i$) to make necessary adjustments. Although many previous papers have proposed similar ideas, most of them focus on binary settings and may therefore adopt different methods [5-7]. This loss function also allows us to encode all information about $t_i$ in $Γ(x_i)$ and $Δ(x_i)$, rather than in $Υ(x_i)$, which aids in making the estimation of $w$ more precise and accurate. Hence, this loss further enhances bias elimination through re-weighting. Secondly, from a practical standpoint, since $Y$ is unknown and is the variable we aim to predict, it is not practical to include a term to measure and minimize the discrepancy loss between two distributions when one of them is unknown. We agree with you using the causal graph structure to justify the $L_{ind}$ may be confusing **and we have already modified that part (in section 3.2 Independent loss, highlighted in blue) to clarify the motivation and aim. Thank you again for your suggestion!**
>
> [1] Wu A, Kuang K, Yuan J, et al. Learning decomposed representation for counterfactual inference[J]. arXiv preprint arXiv:2006.07040, 2020.
> [2] Yao, Liuyi, et al. "A survey on causal inference." ACM Transactions on Knowledge Discovery from Data (TKDD) 15.5 (2021): 1-46.
>
> [3] Kuang, Kun, et al. "Treatment effect estimation with data-driven variable decomposition." Proceedings of the AAAI Conference on Artificial Intelligence. Vol. 31. No. 1. 2017.
>
> [4] Liu, Dugang, et al. "Mitigating confounding bias in recommendation via information bottleneck." Proceedings of the 15th ACM Conference on Recommender Systems. 2021.
>
> [5] Hassanpour, Negar, and Russell Greiner. "Learning disentangled representations for counterfactual regression." International Conference on Learning Representations. 2020.
>
>
> [6] Shalit, Uri, Fredrik D. Johansson, and David Sontag. "Estimating individual treatment effect: generalization bounds and algorithms." International Conference on Machine Learning. PMLR, 2017.
>
> [7] Bellot A, Dhir A, Prando G. Generalization bounds and algorithms for estimating conditional average treatment effect of dosage[J]. arXiv preprint arXiv:2205.14692, 2022.

---

> ### Author Response · Authors · 2023-11-18
> **Rebuttal by Authors**
>
> 3.**Gap between the loss function and the theoretical analysis**
>
> We recognize that there appears to be a gap between the sophisticated loss function and the theoretical analysis. However, we kindly argue that each term in our loss function **contributes meaningfully to the theoretical proof**. Specifically, some components of the loss function serve to support the assumptions underpinning our proof, as outlined in Assumptions 1-4 in our paper. Their functions are introduced as follows. In Assumption 4, we posit that each latent factor should be independent of each other, where discrepancy loss $L_{dis}$ is used to ensure this. $L_{ind}$ enforces us to encode all information of $t_i$ in $\Gamma(x_i)$ and $\Delta(x_i)$ instead of $\Upsilon(x_i)$, thereby facilitating a more precise and accurate estimation of the weight $w$ (re-weighting before the factual loss). Moreover, the weight $w$ before factual loss is the key of the proof, hence we use the $L_T$ loss to get the accurate weight. **Thank you for your suggestion, we have added a paragraph in the Appendix A, highlighted in blue, to clarify the function of each loss in our revised version.**
>
>
> 4.**More results for sensitivity analysis**
>
> We are grateful for your constructive suggestion regarding the addition of more results to enhance the robustness of our experiments. In line with your advice, **we have added an ablation study for $\alpha$ (Table 2) and a sensitivity analysis for $\alpha$ and $\lambda$ based on both MISE and AMSE in our revised paper (Figure 3&6)**. Due to space limitations, we have chosen to only display the figures based on MISE, while the results for AMSE are included in Appendix I.  The results show that despite $\alpha$ also contributes to IDRF estimation, its values have a minimal impact on model performance.  As for $λ$, which controls the regularization, excluding it leads to a decrease in performance. However, when included, the specific value of $λ$ does not significantly affect the results.
>
> 5.**Meaning of $Γ(x_i)$ and $Δ(x_i)$ and clarification on KL divergence**
>
> In our paper, $X$ represents the entire dataset of covariates, encompassing multiple features, rather than a singular random variable. In the previous given example, within the medical historical data of all patients, $X$ represents all features of all patients. For a specific patient $x_i$, the the patient’s age and gender are the confounder factors $Δ(x_i)$, while the patient’s doctor-in-charge is the instrumental factor $Γ(x_i)$. It is apparent that the doctor-in-charge is independent of the patient's age and gender. We believe that adding this example, as per your suggestion, will elucidate this point more clearly. Moreover, $Γ(x_i)$ and $Δ(x_i)$ generated by our model are not random variables; they are deep representations with a dimension of 50, as generated by three representation networks in our model (Figure 2). We agree with your observation that KL divergence is employed to measure the discrepancy between two distributions of representations. In our work, we specifically use the KL divergence loss, with its detailed implementation available in PyTorch: [https://pytorch.org/docs/stable/generated/torch.nn.KLDivLoss.html](https://pytorch.org/docs/stable/generated/torch.nn.KLDivLoss.html). **Your suggestion is valuable. In response, we have added a section in the Appendix D, highlighted in blue, to clarify KL divergence. Additionally, we have included a sentence in the methods section to direct readers to this new Appendix section.**
>
>
> We hope that our response addresses your concerns adequately, and we are committed to incorporating your valuable feedback into the revised version of our paper. Thank you once again for your insightful comments.
>
> [1] Wu A, Kuang K, Yuan J, et al. Learning decomposed representation for counterfactual inference[J]. arXiv preprint arXiv:2006.07040, 2020.
> [2] Yao, Liuyi, et al. "A survey on causal inference." ACM Transactions on Knowledge Discovery from Data (TKDD) 15.5 (2021): 1-46.
>
> [3] Kuang, Kun, et al. "Treatment effect estimation with data-driven variable decomposition." Proceedings of the AAAI Conference on Artificial Intelligence. Vol. 31. No. 1. 2017.
>
> [4] Liu, Dugang, et al. "Mitigating confounding bias in recommendation via information bottleneck." Proceedings of the 15th ACM Conference on Recommender Systems. 2021.
>
> [5] Hassanpour, Negar, and Russell Greiner. "Learning disentangled representations for counterfactual regression." International Conference on Learning Representations. 2020.
>
>
> [6] Shalit, Uri, Fredrik D. Johansson, and David Sontag. "Estimating individual treatment effect: generalization bounds and algorithms." International Conference on Machine Learning. PMLR, 2017.
>
> [7] Bellot A, Dhir A, Prando G. Generalization bounds and algorithms for estimating conditional average treatment effect of dosage[J]. arXiv preprint arXiv:2205.14692, 2022.

---

> > ### Comment · Reviewer_cz1b · 2023-11-20
> > **Response to Authors**
> >
> > Thanks for your time and labor in addressing my concerns. However, I still doubt the soundness of this paper.
> >
> > When we say "A is independent of B", both A and B are typically random variables. However, according to point 5 of your rebuttal, I obtained the following demonstrations.
> >
> > 1. For a specific patient $x_i$, the patient’s age and gender are the confounder factors $\Gamma(x_i)$, while the patient’s doctor-in-charge is the instrumental factor $\Delta(x_i)$.
> >
> > 2. The doctor-in-charge is independent of the patient's age and gender.
> >
> > 3. $\Gamma(x_i)$ and $\Delta(x_i)$ are not random variables. (Maybe they are two values since $x_i$ is a specific patient?)
> >
> > I think it is quite strange to say two values are independent of each other. Please provide further clarification on this point first.

---

> > > ### Author Response · Authors · 2023-11-20
> > > **Further Clarification - continued**
> > >
> > > Dear Reviewer cz1b,
> > >
> > > We would like to further clarify the implementation of the independent loss. In our paper, the intent of the independent loss is to **make the three representations more separate from each other**. We were inspired by the KL divergence and the KL divergence loss in PyTorch to design this loss. After carefully considering your comments and those of Reviewer 8hoc, we agree that referring to it as 'KL divergence' is not appropriate. As you mentioned, KL divergence is typically used to calculate the discrepancy between probability distributions, which is indeed different from ours. Therefore, we have decided to rename this loss as 'divergence loss' $L_D$ and include a full definition of this loss in our revised paper to avoid confusion (the modifications are highlighted in blue). However, we kindly argue that **our loss is still effective for separating the representations**.  for example, $L_D(\Gamma(x_i), \Delta(x_i))$ is calculated as follows :
> > >
> > > $L_{temp}(\Gamma(x_i), \Delta(x_i)) = \Delta(x_i)\log\left(\frac{\Delta(x_i)}{\Gamma(x_i)}\right) = \Delta(x_i)(\log(\Delta(x_i)) - \log(\Gamma(x_i)))$
> > >
> > > $L_D(\Gamma(x_i), \Delta(x_i)) = \frac{1}{n}\sum_{n,m} L_{temp}(\Gamma(x_i), \Delta(x_i)),$
> > >
> > > where $n$ and $m$ represent the batch size and the latent factor dimension, respectively.
> > >
> > > Using this formula, if the representations $\Gamma(x_i)$ and $\Delta(x_i)$ are identical, the result will be 0. Conversely, if $\Gamma(x_i)$ and $\Delta(x_i)$ are distinctly different, the value will be larger. Therefore, we utilize the reverse of this loss to ensure that the representations are **separate** from each other, thereby guaranteeing that each component learns only the information relevant to its specific functionality. To demonstrate the effectiveness of this loss in our model, we present t-SNE plots of the model trained with and without $L_{disc}$ on a synthetic dataset in Fig.2. The incorporation of the discrepancy loss results in more distinct representations and a higher KL-divergence score, validating our loss approach.
> > >
> > > We hope this address your concerns.

---

> ### Author Response · Authors · 2023-11-20
> **Further Clarifications**
>
> Hi Reviewer cz1b,
>
> Thank you for your response. We are happy to provide further clarifications as follows.
>
> To elaborate, $\Gamma(x_i)$ and $\Delta(x_i)$ for a specific patient are not singular values; rather, they are combinations of features specific to that patient. For instance, in this example, age and gender (two features) are the confounder variables for the patient, so their combination forms the confounder factors. In real datasets, the complexity of these combinations is often much larger than in this example. For example, if we have a dataset with 50 features, where 20 are instrumental features, 10 are confounder features, and 20 are adjustment features, then the confounder factors for a specific patient would be the combination of those 10 features. However, extracting and representing these combinations for each patient is challenging, which is why we use a neural network in our model. As illustrated in Fig.1, our network learns three separate representations (each with a dimension of 50, representing one factor) from the input. The independent loss is implemented to ensure that each component learns only the information relevant to its specific functionality.
>
> We hope this response clarifies your question. Should you have any further queries or need additional information, please do not hesitate to let us know.

---

> ### Comment · Reviewer_cz1b · 2023-11-21
> **Further Response to Authors**
>
> Dear authors,
>
> Please answer my following questions **directly** and **briefly**.
>
> 1. You seem to use "independence", "disentanglement", and "separation" exchangeably, please clarify their differences.
>
> 2. If you say "A is independent of B", what is the definition of "independent" if A and B are not random variables?
>
> 3. You say that you want to separate $\Gamma(x_i)$ from $\Delta(x_i)$, but separation does not make sense. Consider A is an n-dim binary vector, and B is XOR(A). Then the divergence between A and B is maximized, but they are not independent of each other.

---

> > ### Author Response · Authors · 2023-11-21
> > **Further Clarifications**
> >
> > Dear reviewer,
> >
> > We apologize for any confusion.
> >
> > 1. In our latest revised paper, we do not claim independence; instead, the goal of the discrepancy loss is to **separate and disentangle**. We recognize that the use of 'KL-divergence' and 'independence' might cause confusion, as previously rebuttal illustrated, so we have already removed them. The terms 'separation' and 'disentanglement' are the **same** because they share the same goal —making the representations separable (separation) to enable them to encode only relevant information (disentanglement). Moreover, in the context of latent space, separation implies disentanglement. We apologize for any confusion caused by the use of 'independence.' The concept of 'independence' is no longer associated with the discrepancy loss in our revised paper.
> >
> > 2. Since 'independence' is deleted from our revised paper, there is no need for a definition of independence in this context.
> >
> > 3. Regarding the example you mentioned, A and B represent exactly what we are aiming for. Although they are related, they are separate from each other. In our case, they are the latent vectors in latent space. Significant separation in latent space indicates that they are disentangled and possess different meanings.
> >
> > We hope this addresses your concerns.

---

> ### Comment · Reviewer_cz1b · 2023-11-22
> **Response to authors**
>
> Dear authors,
>
> In my humble opinion, **separation itself does not make sense**. According to the Markov property, a graph model (Figure 1(a)) entails independence rather than separation. If I'm wrong, please provide some references about "separation derived from graph".

---

> > ### Author Response · Authors · 2023-11-22
> > **Clarifications on Figure 1 a**
> >
> > Dear Reviewer,
> >
> > We would like to clarify our Figure 1a.
> >
> > We kindly argue that Figure 1a is a schematic diagram rather than a strict graph model. It is used to illustrate how the input can be separated into three factors and how the combination of certain specific factors can be utilized to predict $T$ and $Y$. This type of diagram is commonly used in causal inference papers [1-5], and typically, these papers do not assert the independence of these factors, nor do they employ any loss function to ensure it. Moreover, these papers do not include our type of discrepancy loss, which serves to make the three factors more distinct. Without this loss, the representations of the three factors might become entangled in the latent space, potentially leading to inaccurate estimations. Hence, our objective is to accurately encode the representations of the three different factors by promoting their separation in the latent space and to estimate the IDRF based on these representations.
> >
> > We hope this addresses your concern.
> >
> > [1] Wu A, Kuang K, Yuan J, et al. Learning decomposed representation for counterfactual inference[J]. arXiv preprint arXiv:2006.07040, 2020.
> > [2] Yao, Liuyi, et al. "A survey on causal inference." ACM Transactions on Knowledge Discovery from Data (TKDD) 15.5 (2021): 1-46.
> >
> > [3] Kuang, Kun, et al. "Treatment effect estimation with data-driven variable decomposition." Proceedings of the AAAI Conference on Artificial Intelligence. Vol. 31. No. 1. 2017.
> >
> > [4] Liu, Dugang, et al. "Mitigating confounding bias in recommendation via information bottleneck." Proceedings of the 15th ACM Conference on Recommender Systems. 2021.
> >
> > [5] Hassanpour, Negar, and Russell Greiner. "Learning disentangled representations for counterfactual regression." International Conference on Learning Representations. 2020.

---

### Official Review · Reviewer_Wac2 · 2023-11-08

**Soundness:** 3 good
**Presentation:** 3 good
**Contribution:** 2 fair
**Rating:** 6
**Confidence:** 3

**Summary:**

This paper aims to estimate the individual treatment effect (ITE) under continuous treatment setting. The authors claim that the latent covariates can be divided into three different kinds of variables, i.e., instrumental variable, confounder and adjustment variable, and the representations of confounders should be balanced across treatments.
The proposed DBRNet can address the above issues under theoretical guarantees. Extensive experiments are conducted to verify the effectiveness of DBRNet.

**Strengths:**

1.	The paper has a clear goal to estimate individual treatment effects in a continuous treatment setting.
2.	The paper introduces DBRNet as a solution to tackle the estimation challenges, and it comes with theoretical guarantees, indicating that it's a reliable approach.
3.	The paper conducts extensive experiments to show that DBRNet works effectively in practice.

**Weaknesses:**

1.	The motivation of this paper is convincing but lacks innovation because neither the disentangled representation nor the re-weighting technique is originally proposed in this paper. I hope the authors can provide deeper insights into why disentangled covariates should be considered in the context of continuous treatment settings.
2.	In equation (5), why isn't there an enforcement of the discrepancy between the adjustment variable and the treatment variable? This point should be clarified.
3.	In the experimental setting, it would be beneficial to compare the proposed method with more disentanglement-based baselines, such as DR-CFR[1] and DeR-CFR[2].
[1] [ICLR'20] Learning Disentangled Representations for CounterFactual Regression
[2] [ArXiv'20] Learning decomposed representation for counterfactual inference

**Questions:**

see the above comments

---

> ### Author Response · Authors · 2023-11-18
> **Rebuttal by Authors**
>
> To Reviewer Wac2:
>
> 1.**Motivation of using disentanglement in the context of continuous settings**
>
> We are grateful for your question. The motivation for using disentanglement in continuous settings is to **precisely** adjust selection bias. Existing methods often resort to balancing the entire representation, a simple yet brute approach. However, we argue that not all information in the latent representation should be balanced to adjust for selection bias. For instance, while confounder factors in the representation bring selection bias, they also contribute to outcome predictions. Balancing instrumental factors is theoretically implausible since they relate to treatment assignment and should not be identical across treatments. Therefore, indiscriminate balancing of the entire representation of input covariates is not appropriate. Hence, we apply disentanglement techniques to disentangle the three factors for the further precise bias adjustment. **We appreciate your suggestion and have added more details about the motivation (highlighted in blue) in both the introduction and the methods sections of our paper.**
>
> 2.**Clarification on KL divergence**
>
> Thank you for prompting us to clarify our approach to KL divergence. In our model, both instrumental $\Gamma(x_i)$ and confounder representations $\Delta(x_i)$ are used to estimate the probability of treatment, while confounder $\Delta(x_i)$ and adjustment representations $\Upsilon(x_i)$ are used to predict the outcome. We argue that instrumental and confounder representations are more similar, as are confounder and adjustment representations, since they are optimized for the same goals. Therefore, we only penalize these. To avoid making our model cumbersome, we decided not to include the KL divergence between instrumental $\Gamma(x_i)$ and adjustment representations $\Upsilon(x_i)$ . **We have added a section in the Appendix D, highlighted in blue, to clarify KL divergence. Additionally, we have included a sentence in the methods section to direct readers to this new Appendix section.**
>
> 3.**Comparison with other methods**
>
> Thank you for sharing these interesting papers! After a careful review, one of them [1] is extensively discussed in the related work and section 3.4 of our paper. The limitations of [1] actually motivated our approach. Specifically, [1] suggests building regression networks on the top of disentangled embeddings for each treatment value in binary settings. However, in continuous settings, building a network for each treatment value is impractical due to the infinite nature of $T$. Moreover, upon revisiting their code according to your suggestion, we note that [1] requires prior knowledge of the probability of each treatment to compute the re-weighting function, making it unsuitable for continuous treatment scenarios. Similarly, the methods in [2] are limited, as they binarize the treatment. The lack of open-source code for [2] further complicates direct comparison. Overall, while these methods employ disentanglement, they address different questions in different settings. We would be eager to compare our method with theirs if they release their code, especially for continuous settings.
>
> We hope that our response adequately addresses your concerns. We are committed to incorporating your valuable feedback into the revised version of our paper. Thank you once again for your insightful comments.
>
>
> [1] Hassanpour N, Greiner R. Learning disentangled representations for counterfactual regression[C]//International Conference on Learning Representations. 2019.
>
> [2] Wu A, Kuang K, Yuan J, et al. Learning decomposed representation for counterfactual inference[J]. arXiv preprint arXiv:2006.07040, 2020.

---

> > ### Comment · Reviewer_Wac2 · 2023-11-21
> > **response**
> >
> > Thanks for the authors' rebuttal. I'd like to keep my already positive score

---

### Author Response · Authors · 2023-11-18
**Summary of Paper Modifications**

Dear Reviewers,

Thank you very much for your constructive suggestions. We respectfully acknowledge and value your input, and have made several additions and modifications to our paper in response. To address your major concerns, we have conducted additional experiments and updated the manuscript, including:
- Further ablation studies for $\alpha$ (Table 2)
- Expanded sensitivity testing for all hyperparameters based on both AMSE and MISE (Figure 3 & Figure 6).
- Additional explanations and an example of our causal graph (in Appendix H) for enhanced understanding.
- More detailed explanations of our proof, including the functionality of every term in our loss function, along with supplemental proofs for further clarification (Appendix A).
- Clarifications regarding the KL divergence loss and its implementation (Appendix D).
- Expanded descriptions and clarifications of certain components of our model to prevent potential confusion.
- Corrections of typographical errors as pointed out by the reviewers.

---

### Meta-Review · Area_Chair_awj6 · 2023-12-08

**Metareview:**

This paper focuses on the problem of estimating the individual-level continuous treatment effect by learning a disentangled representation.

During the rebuttal, reviewers acknowledged that: 1) it contributes to an important problem; 2) theoretical justifications are provided; and 3) the overall written quality is high.

However,  during the discussion, one central concern was also raised about the correctness of the method for learning disentangled representation. Specifically, both Reviewer 8hoc and Reviewer cz1b pointed out that using a divergence loss on top of the representations does not guarantee the representations to be disentangled.

We believe the paper would be strong after addressing this concern. In its current form, the AC regretfully rejects it.

**Justification For Why Not Higher Score:**

Overall, this is an interesting paper. The concerns regarding using a divergence loss to disentangle representations have been intensively discussed. We believe that the paper would be a strong one by solving the concerns in the next round.

**Justification For Why Not Lower Score:**

NA

---

### Decision · Program_Chairs · 2024-01-16

Reject